# A3FL: Adversarially Adaptive Backdoor Attacks to Federated Learning

**Hangfan Zhang, Jinyuan Jia, Jinghui Chen, Lu Lin, Dinghao Wu**
{hbz5148,jinyuan,jzc5917,lulin,dinghao}@psu.edu
The Pennsylvania State University

## Abstract

Federated Learning (FL) is a distributed machine learning paradigm that allows multiple clients to train a global model collaboratively without sharing their local training data. Due to its distributed nature, many studies have shown that it is vulnerable to backdoor attacks. However, existing studies usually used a predetermined, fixed backdoor trigger or optimized it based solely on the local data and model without considering the global training dynamics. This leads to sub-optimal and less durable attack effectiveness, i.e., their attack success rate is low when the attack budget is limited and decreases quickly if the attacker can no longer perform attacks anymore. To address these limitations, we propose A3FL, a new backdoor attack which adversarially adapts the backdoor trigger to make it less likely to be removed by the global training dynamics. Our key intuition is that the difference between the global model and the local model in FL makes the local-optimized trigger much less effective when transferred to the global model. We solve this by optimizing the trigger to even survive the scenario where the global model was trained to directly unlearn the trigger. Extensive experiments on benchmark datasets are conducted for twelve existing defenses to comprehensively evaluate the effectiveness of our A3FL. Our code is available at https://github.com/hfzhang31/A3FL.

## 1 Introduction

Recent years have witnessed the rapid development of Federated Learning (FL) [1, 2, 3], an advanced distributed learning paradigm. With the assistance of a cloud server, multiple clients such as smartphones or IoT devices train a global model collaboratively based on their private training data through multiple communication rounds. In each communication round, the cloud server selects a part of the clients and sends the current global model to them. Each selected client first uses the received global model to initialize its local model, then trains it based on its local dataset, and finally sends the trained local model back to the cloud server. The cloud server aggregates local models from selected clients to update the current global model. FL has been widely used in many safety- and privacy-critical applications [4, 5, 6, 7].

Numerous studies [8, 9, 10, 11, 12, 13, 14] have shown that the distributed nature of FL provides a surface to backdoor attacks, where an attacker can compromise some clients and utilize them to inject a backdoor into the global model such that the model's behaviors are the attacker desired. In particular, the backdoored global model behaves normally on clean testing inputs but predicts any testing inputs stamped with an attacker-chosen backdoor trigger as a specific target class.

Depending on whether the backdoor trigger is optimized, we can categorize existing attacks into *fixed-trigger attacks* [12, 11, 13, 8] and *trigger-optimization attacks* [10, 9]. In a fixed-trigger attack, an attacker pre-selects a fixed backdoor trigger and thus does not utilize any information from FL training process. CWhile a fixed-trigger attack can be more efficient and straightforward, it usually suffers from limited effectiveness and more obvious utility drops.

In a trigger-optimization attack, an attacker optimizes the backdoor trigger to enhance the attack. Fang et al. [10] proposed to maximize the difference between latent representations of clean and trigger-stamped samples. Lyu et al. [9] proposed to optimize the trigger and local model jointly with $\ell_2$ regularization on local model weights to bypass defenses. The major limitations of existing trigger-optimization attacks are twofold. First, they only leverage local models of compromised clients to optimize the backdoor trigger, which ignores the global training dynamics. Second, they strictly regulate the difference between the local and global model weights to bypass defenses, which in turn limits the backdoor effectiveness. As a result, the locally optimized trigger becomes much less effective when transferred to the global model as visualized in Figure 1. More details for this experiment can be found in Appendix A.1.

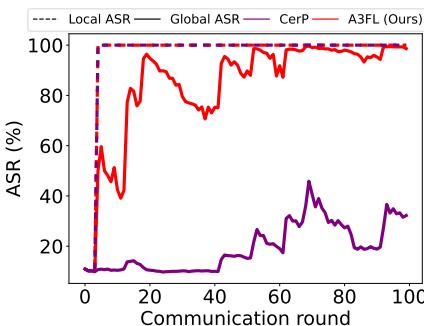

Figure 1: A3FL and CerP [9] can achieve 100% ASR on the local model. However, only A3FL in the mean time obtains a high global ASR.

**Our contribution:** In this paper, we propose **A**dversarially **A**daptive Backdoor **A**ttacks to **F**ederated **L**earning (A3FL). Recall that existing works can only achieve sub-optimal attack performance due to ignorance of global training dynamics. A3FL addresses this problem by adversarially adapting to the dynamic global model. We propose *adversarial adaptation loss*, in which we apply an adversarial training-like method to optimize the backdoor trigger so that the injected backdoor can remain effective in the global model. In particular, we predict the movement of the global model by assuming that the server can access the backdoor trigger and train the global model to directly unlearn the trigger. We adaptively optimize the backdoor trigger to make it survive this adversarial global model, i.e., the backdoor cannot be easily unlearned even if the server is aware of the exact backdoor trigger. We empirically validate our intuition as well as the effectiveness and durability of the proposed attack.

We further conduct extensive experiments on widely-used benchmark datasets, including CIFAR-10 [15] and TinyImageNet [16], to evaluate the effectiveness of A3FL. Our empirical results demonstrate that A3FL is consistently effective across different datasets and settings. We further compare A3FL with 4 state-of-the-art backdoor attacks [12, 11, 10, 9] under 13 defenses [2, 17, 18, 19, 20, 21, 22, 23, 24, 25, 26, 27], and the results suggest that A3FL remarkably outperforms all baseline attacks by up to 10 times against all defenses. In addition, we find that A3FL is significantly more durable than all baselines. Finally, we conduct extensive ablation studies to evaluate the impact of hyperparameters on the performance of A3FL.

To summarize, our contributions can be outlined as follows.

- We propose A3FL, a novel backdoor attack to the FL paradigm based on adversarial adaptation, in which the attacker optimizes the backdoor trigger using an adversarial training-like technique to enhance its persistence within the global training dynamics.

- We empirically demonstrate that A3FL remarkably improves the durability and attack effectiveness of the injected backdoor in comparison to previous backdoor attacks.

- We comprehensively evaluate A3FL towards existing defenses and show that they are insufficient for mitigating A3FL, highlighting the need for new defenses.

## 2 Related Work

**Federated learning:** Federated Learning (FL) was first proposed in [1] to improve communication efficiency in decentralized learning. FedAvg [2] aggregated updates from each client and trains the global model with SGD. Following studies [28, 29, 30, 31, 32] further improved the federated paradigm by making it more adaptive, general, and efficient.

**Existing attacks and their limitations:** In backdoor attacks to FL, an attacker aims to inject a backdoor into model updates of compromised clients such that the final global model aggregated by the server is backdoored. Existing backdoor attacks on FL can be classified into two categories: fixed-trigger attacks [12, 11, 8, 14, 13] and trigger-optimization attacks [10, 9].

Fixed-trigger attacks [8, 11, 14, 13, 12] pre-select a fixed backdoor trigger and poison the local training set with it. Since a fixed trigger may not be effective for backdoor injection, these attacks improved the backdoor effectiveness through other approaches including manually manipulating the poisoned updates. Particularly, scaling attack [8] scaled up the updates to dominate other clients to improve the attack effectiveness. DBA [11] split the trigger into several sub-triggers for poisoning, which makes DBA more stealthy from defenses. Neurotoxin [12] only attacked unimportant model parameters that are less frequently updated to prevent the backdoor from being erased shortly.

Trigger-optimization attacks [10, 9] optimize the backdoor trigger to enhance the attack. F3BA [10] optimized the trigger pattern to maximize the difference between latent representations of clean and trigger-stamped samples. F3BA also projected gradients to unimportant model parameters like Neurotoxin [12] to improve stealthiness. CerP [9] jointly optimized the trigger and the model weights with $\ell_2$ regularization to minimize the local model bias. These attacks can achieve higher attack performance than fixed-trigger attacks. However, they have the following limitations. First, they only consider the static local model and ignore the dynamic global model in FL, thus the optimized trigger could be sub-optimal on the global model. Second, they apply strict regularization on the difference between the local model and the global model, which harms the backdoor effectiveness. Therefore, they commonly need a larger attack budget (e.g., compromising more clients) to take effect. We will empirically demonstrate these limitations in Section 4.

**Existing defenses:** In this paper, we consider two categories of defenses in FL. The first category of defense mechanisms is deliberately designed to alleviate the risks of backdoor attacks [17, 19, 20, 18, 33] on FL. These defense strategies work by restricting clients' updates to prevent the attackers from effectively implanting a backdoor into the global model. For instance, the Norm Clipping [17] defense mechanism limits clients' behavior by clipping large updates, while the CRFL [19] defense mechanism uses parameter smoothing to impose further constraints on clients' updates.

The second category of defenses [26, 25, 24, 23, 22, 21, 34] is proposed to improve the robustness of FL against varied threats. These defense mechanisms operate under the assumption that the behavior of different clients is comparable. Therefore, they exclude abnormal clients to obtain an update that is recognized by most clients to train the global model. For instance, the Median [22] defense mechanism updates the global model using the median values of all clients' updates, while Krum [21] filters out the client with the smallest pairwise distance from other clients and trains the global model solely with the filtered client updates. These defense mechanisms can achieve superior robustness compared to those defense mechanisms that are specifically designed for backdoor attacks. Nevertheless, the drawback of this approach is evident: it often compromises the accuracy of the global model, as it tends to discard most of the information provided by clients, even if these updates are merely potentially harmful.

In this paper, we also utilize backdoor unlearning [35, 36] to approximate existing defenses. Backdoor unlearning typically minimizes the prediction loss of trigger-stamped samples to the ground truth labels. Note that backdoor unlearning disparts from so-called machine unlearning [37, 38, 39], in which the model is unlearned to "forget" specific training samples.

There exist additional defenses in FL that are beyond the scope of this paper. While these defenses may offer potential benefits, they also come with certain limitations in practice. For instance, FLTrust [40] assumed the server holds a clean validation dataset, which deviates from the typical FL setting. Cao et al. [41] proposed sample-wise certified robustness which demands hundreds of times of retraining and is computationally expensive.

## 3 Methodology

To formulate the backdoor attack scenario, we first introduce the federated learning setup and threat model. Motivated by the observation of the local-global gap in existing works due to the ignorance of global dynamics, we propose to optimize the trigger via an adversarial adaptation loss.

### 3.1 Federated Learning Setup and Threat Model

We consider a standard federated learning setup where $N$ clients aim to collaboratively train a global model $f$ with the coordination of a server. Let $\mathcal{D}_i$ be the private training dataset held by the client $i$, where $i = 1, 2, \ldots, N$. The joint training dataset of the $N$ clients can be denoted as $\mathcal{D} = \cup_{i=1}^{N} \mathcal{D}_i$.

In the $t$-th communication round, the server first randomly selects $M$ clients, where $M \leq N$. For simplicity, we use $S_t$ to denote the set of selected $M$ clients. The server then distributes the current version of the global model $\boldsymbol{\theta}_t$ to the selected clients. Each selected client $i \in S_t$ first uses the global model to initialize its local model, then trains its local model on its local training dataset, and finally uploads the local model update (i.e., the difference between the trained local model and the received global model) to the server. We use $\boldsymbol{\Delta}_t^i$ to denote the local model update of the client $i$ in the $t$-th communication round. The server aggregates the received updates on model weights and updates the current global model weights as follows:

$$\boldsymbol{\theta}_{t+1} = \boldsymbol{\theta}_t + \mathcal{A}(\{\boldsymbol{\Delta}_t^i | i \in S_t\}) \tag{1}$$

where $\mathcal{A}$ is an aggregation rule adopted by the server. For instance, a widely used aggregation rule FedAvg [2] takes an average over the local model updates uploaded by clients.

**Attacker's goal:** We consider an attacker aims to inject a backdoor into the global model. In particular, the attacker aims to make the injected backdoor **effective** and **durable**. The backdoor is **effective** if the backdoored global model predicts any testing inputs stamped with an attacker-chosen backdoor trigger as an attacker-chosen target class. The backdoor is **durable** if it remains in the global model even if the attacker-compromised clients stop uploading poisoned updates while the training of the global model continues. We note that a durable backdoor is essential for an attacker as the global model in a production federated learning system is periodically updated but it is impractical for the attacker to perform attacks in all time periods [12, 42]. Considering the durability of the backdoor enables us to understand the effectiveness of backdoor attacks under a strong constraint, i.e., the attacker can only attack the global model within a limited number of communication rounds.

**Attacker's background knowledge and capability:** Following threat models in previous studies [9, 12, 10, 11, 14, 8], we consider an attacker that can compromise a certain number of clients. In particular, the attacker can access the training datasets of those compromised clients. Moreover, the attacker can access the global model received by those clients and manipulate their uploaded updates to the server. As a practical matter, we consider the attacker can only control those compromised clients for a limited number of communication rounds [12, 11, 8, 10, 9].

### 3.2 Adversarially Adaptive Backdoor Attack (A3FL)

Our key observation is that existing backdoor attacks are less effective because they either use a fixed trigger pattern or optimize the trigger pattern only based on the local model of compromised clients. However, the global model is dynamically updated and therefore differs from the static local models. This poses two significant challenges for existing backdoor attacks. Firstly, a backdoor that works effectively on the local model may not be similarly effective on the global model. Secondly, the injected backdoor is rapidly eliminated since the global model is continuously updated by the server, making it challenging for attackers to maintain the backdoor's effectiveness over time.

We aim to address these challenges by adversarially adapting the backdoor trigger to make it persistent in the global training dynamics. Our primary objective is to optimize the backdoor trigger in a way that allows it to survive even in the scenario where the global model is trained to directly unlearn the backdoor [35, 36]. To better motivate our method, we first discuss the limitations of existing state-of-the-art backdoor attacks on federated learning.

**Limitation of existing works:** In recent state-of-the-art works [9, 10], the attacker optimizes the backdoor trigger to maximize its attack effectiveness and applies regularization techniques to bypass server-side defense mechanisms. Formally, given the trigger pattern $\boldsymbol{\delta}$ and an arbitrary input $\mathbf{x}$, the input stamped with the backdoor trigger can be denoted as $\mathbf{x} \oplus \boldsymbol{\delta}$, which is called the backdoored input. Suppose the target class is $\tilde{y}$. Since the attacker has access to the training dataset of a compromised client $i$, the backdoor trigger $\boldsymbol{\delta}$ can be optimized using the following objective:

$$\boldsymbol{\delta}^* = \underset{\boldsymbol{\delta}}{\arg\min} \, \mathbb{E}_{(\mathbf{x},y) \sim \mathcal{D}_i} \left[ \mathcal{L}(\mathbf{x} \oplus \boldsymbol{\delta}, \tilde{y}; \boldsymbol{\theta}_t) \right] \tag{2}$$

where $\boldsymbol{\theta}_t$ represents the global model weights in the $t$-th communication round, and $\mathcal{L}$ is the classification loss function such as cross-entropy loss. To conduct a backdoor attack locally, the attacker randomly samples a small set of inputs $\mathcal{D}_i^b$ from the local training set $\mathcal{D}_i$, and poisons inputs in $\mathcal{D}_i^b$ with trigger stamped. The attacker then injects a backdoor into the local model by optimizing the local model on the partially poisoned local training set with regularization to limit the gap between

the local and global model, i.e., $||\boldsymbol{\theta} - \boldsymbol{\theta}_t||$. While the regularization term helps bypass server-side defenses, it greatly limits the backdoor effectiveness, as it only considers the current global model $\boldsymbol{\theta}_t$ and thus fails to adapt to future global updates.

As illustrated in Figure 1, we observe that such backdoor attack on federated learning (e.g., CerP [9]) is highly effective on the local model, suggested by a high local attack success rate (ASR). However, due to the ignorance of global dynamics, they cannot achieve similar effectiveness when transferred to the global model, resulting in a low ASR on the global model. Our method A3FL aims to bridge the local-global gap in existing approaches to make the backdoor persistent when transferred to the global model thus achieving advanced attack performance. In particular, we introduce *adversarial adaptation loss* that makes the backdoor persistent to global training dynamics.

**Adversarial adaptation loss:** To address the challenge introduced by the global model dynamics in federated learning, we propose the *adversarial adaptation loss*. As the attacker cannot directly control how the global model is updated as federated learning proceeds, its backdoor performance can be significantly impacted when transferred to the global model, especially when only a small number of clients are compromised by the attacker or defense strategies are deployed. For instance, local model updates from benign clients can re-calibrate the global model to indirectly mitigate the influence of the backdoored updates from the compromised clients; a defense strategy can also be deployed by the server to mitigate the backdoor. To make the backdoor survive such challenging scenarios, our intuition is that, if an attacker could anticipate the future dynamics of the global model, the backdoor trigger would be better optimized to adapt to global dynamics.

However, global model dynamics are hard to predict because 1) at each communication round, all selected clients contribute to the global model but the attacker cannot access the private training datasets from benign clients and thus cannot predict their local model updates, and 2) the attacker does not know how local model updates are aggregated to obtain the global model and is not aware of possible defense strategies adopted by the server. As directly predicting the exact global model dynamics is challenging, we instead require the attacker to foresee and survive the scenario where the global model is trained to directly unlearn the backdoor. In this paper we consider backdoor unlearning proposed in prior backdoor defenses [35, 36].

Specifically, starting from current global model $\boldsymbol{\theta}_t$, we foresee an adversarially crafted global model $\boldsymbol{\theta}_t'$ that can minimize the impact of the backdoor. We adopt an adversarial training-like method to obtain $\boldsymbol{\theta}_t'$: the attacker can use the generated backdoor trigger to simulate the unlearning of the backdoor in the global model. The trigger is then optimized to simultaneously backdoor the current global model $\boldsymbol{\theta}_t$ and the adversarially adapted global model $\boldsymbol{\theta}_t'$. Formally, the adversarially adaptive backdoor attack (A3FL) can be formulated as the following optimization problem:

$$
\begin{aligned}
\boldsymbol{\delta}^* &= \operatorname*{argmin}_{\boldsymbol{\delta}} \mathbb{E}_{(\mathbf{x},y)\sim\mathcal{D}_i} \big[ \mathcal{L}(\mathbf{x} \oplus \boldsymbol{\delta}, \tilde{y}; \boldsymbol{\theta}_t) + \lambda \mathcal{L}(\mathbf{x} \oplus \boldsymbol{\delta}, \tilde{y}; \boldsymbol{\theta}_t') \big] \\
s.t.\ \boldsymbol{\theta}_t' &= \operatorname*{argmin}_{\boldsymbol{\theta}} \mathbb{E}_{(\mathbf{x},y)\sim\mathcal{D}_i} \big[ \mathcal{L}(\mathbf{x} \oplus \boldsymbol{\delta}, y; \boldsymbol{\theta}) \big]
\end{aligned}
\tag{3}
$$

where $\boldsymbol{\theta}$ is initialized with current global model weights $\boldsymbol{\theta}_t$; $\boldsymbol{\theta}_t'$ is the optimized adversarial global model which aims to correctly classify the backdoored inputs as their ground-truth label to unlearn the backdoor. In trigger optimization, $\lambda$ is a hyperparameter balancing the backdoor effect on the current global model $\boldsymbol{\theta}_t$ and the adversarial one $\boldsymbol{\theta}_t'$, such that the local-global gap is bridged when the locally optimized trigger is transferred to the global model (after server-side aggregation/defenses). Note that attacking the adversarial model is an adaptation or approximation of global dynamics, as in practice the server cannot directly access and unlearn the backdoor trigger to obtain such an adversarial model.

**Algorithm of A3FL:** We depict the workflow of A3FL compromising a client in Algorithm 1. At the $t$-th communication round, the client is selected by the server and receives the current global model $\boldsymbol{\theta}_t$. Lines 4-8 optimize the trigger based on the current and the adversarial global model using cross-entropy loss $\mathcal{L}_{ce}$. The adversarial global model is initialized by the global model weights in Line 1, and is updated in Line 10. Lines 12-14 train the local model on the poisoned dataset and upload local updates to the server.

**Algorithm 1:** The workflow of A3FL compromising a client

---

**Input:** $\boldsymbol{\theta}_t, \mathcal{D}_i, \tilde{y}, K, K_{\text{trigger}}, \alpha_1, \alpha_2, \lambda$

 1: $\boldsymbol{\theta}'_t = \boldsymbol{\theta}_t$
 2: **for** $j = 1$ to $K$ **do**
 3:     Sample a batch of training data $\mathcal{B}$ from $\mathcal{D}_i$
 4:     **for** $k = 1$ to $K_{\text{trigger}}$ **do**
 5:       // Optimize trigger pattern $\boldsymbol{\delta}$ following Equation 3.
 6:       $L = \frac{1}{|\mathcal{B}|} \sum_{\mathbf{x} \in \mathcal{B}} (\mathcal{L}_{\text{ce}}(\mathbf{x} \oplus \boldsymbol{\delta}, \tilde{y}; \boldsymbol{\theta}_t) + \lambda \mathcal{L}_{\text{ce}}(\mathbf{x} \oplus \boldsymbol{\delta}, \tilde{y}; \boldsymbol{\theta}'_t))$
 7:       $\boldsymbol{\delta} \leftarrow \boldsymbol{\delta} - \alpha_1 \nabla_{\boldsymbol{\delta}} L$
 8:     **end for**
 9:     // Optimize adversarial global model weights $\boldsymbol{\theta}'_t$ following Equation 3.
10:     $\boldsymbol{\theta}'_t \leftarrow \boldsymbol{\theta}'_t - \alpha_2 \nabla_{\boldsymbol{\theta}} \frac{1}{|\mathcal{B}|} \sum_{(\mathbf{x},y) \in \mathcal{B}} \mathcal{L}_{\text{ce}}(\mathbf{x} \oplus \boldsymbol{\delta}, y; \boldsymbol{\theta}'_t)$
11: **end for**
12: Poison local dataset with $\boldsymbol{\delta}$ and update local model to obtain $\boldsymbol{\theta}^i_{t+1}$
13: $\boldsymbol{\Delta}^{t+1}_i = \boldsymbol{\theta}^i_{t+1} - \boldsymbol{\theta}_t$
14: Upload $\boldsymbol{\Delta}^{t+1}_i$ to the server

---

## 4 Experiments

### 4.1 Experimental Setup

**Datasets:** We evaluate A3FL on three widely-used benchmark datasets: FEMNIST [43], CIFAR-10 [15], and TinyImageNet [16]. The FEMNIST dataset consists of 80,5263 images shaped $28 \times 28$ distributed across 10 classes. The CIFAR-10 dataset consists of 50,000 training images and 10,000 testing images that are uniformly distributed across 10 classes, with each image having a size of $32 \times 32$ pixels. The TinyImageNet dataset contains 100,000 training images and 20,000 testing images that are uniformly distributed across 200 classes, where each image has a size of $64 \times 64$ pixels.

**Federated learning setup:** By default, we set the number of clients $N = 100$. At each communication round, the server randomly selects $M = 10$ clients to contribute to the global model. The global model architecture is ResNet-18 [44]. We assume a non-i.i.d data distribution with a concentration parameter $h$ of 0.9 following previous works [12, 10, 9]. We evaluate the impact of data heterogeneity by adjusting the value of $h$ in Appendix B.6. Each selected client trains the local model for 2 epochs using SGD optimizer with a learning rate of 0.01. The FL training process continues for 2,000 communication rounds.

**Attack setup:** We assume that the attacker compromises $P$ clients among all $N$ clients. All compromised clients are only allowed to attack in limited communication rounds called *attack window*. By default, the attack window starts at the 1,900th communication round and ends at the 2,000th communication round. We also discuss the impact of the attack window in Appendix B.7. When a compromised client is selected by the server during the attack window, it will upload poisoned updates trying to inject the backdoor. We adjust the number of compromised clients $P \in [1, 20]$ to comprehensively evaluate the performance of each attack. Different from previous works [11, 12], in our evaluation compromised clients are selected randomly to simulate the practical scenario. Each compromised client poisons 25% of the local training dataset and trains the local model on the partially poisoned dataset with the same parameter settings as benign clients unless otherwise mentioned. By default, the trigger is designed as a square at the upper left corner of the input images. We use the same trigger design for all baseline attacks to ensure the same level of data stealthiness for a fair comparison. We summarize the details of each attack in Appendix A.2. We also discuss different trigger designs of DBA [11] in Appendix B.9.

**A3FL setup:** By default, compromised clients optimize the trigger using Projected Gradient Descent (PGD) [45] with a step size of 0.01. The adversarial global model is optimized using SGD with a learning rate of 0.01. In practice, we set the balancing coefficient $\lambda = \lambda_0 \text{sim}(\boldsymbol{\theta}'_t, \boldsymbol{\theta}_t)$, where $\text{sim}(\boldsymbol{\theta}'_t, \boldsymbol{\theta}_t)$ denotes the cosine similarity between $\boldsymbol{\theta}'_t$ and $\boldsymbol{\theta}_t$. We use similarity to automatically adjust the focus to the adversarial global model: if the adversarial global model is similar to the current global model, it will be assigned a higher weight; otherwise, the adversarial global model is assigned

Table 1: A3FL maintains the utility of the global model on CIFAR-10.

| Defense | FedAvg | NC | RLR | Median | DSight | Bulyan | Krum | SFed | CRFL | DP | FedDF | FedRAD |
|---------|--------|-------|-------|--------|--------|--------|-------|-------|-------|-------|-------|--------|
| ACC(%) | 92.29 | 92.57 | 92.21 | 65.59 | 91.79 | 39.57 | 84.56 | 92.60 | 87.40 | 87.71 | 37.58 | 65.89 |
| BAC(%) | 92.44 | 92.61 | 92.26 | 65.53 | 91.79 | 39.92 | 84.41 | 92.70 | 87.35 | 87.60 | 40.09 | 65.61 |

a lower weight. We use the similarity to control the strength of adversarial training, since the backdoor could be fully unlearned if the adversarial global model is aggressively optimized, which makes it difficult to optimize the first term in Equation 3. In adversarial scenarios, it is important to balance the strengths of both sides to achieve better performance, which has been well studied in previous works in adversarial generation [46, 47]. When there are multiple compromised clients in $S_t$, the backdoor trigger is optimized on one randomly selected compromised client, and all compromised clients use this same trigger. We also discuss the parameter setting of A3FL in experiments.

**Compared attack baselines:** We compare our A3FL to four representative or state-of-the-art backdoor attacks to FL: Neurotoxin [12], DBA [11], CerP [9], and F3BA [10]. We discuss these baselines in Section 2 and also provide an in-detail introduction in Appendix A.2 including specific hyperparameter settings and trigger design of each baseline.

**Compared defense baselines:** We evaluate A3FL under 13 state-of-the-art or representative federated learning defenses: FedAvg [2], Median [22], Norm Clipping [17], DP [17], Robust Learning Rate [18], Deepsight [20], Bulyan [23], FedDF [24], FedRAD [25], Krum [21], SparseFed [26], FLAME [27], and CRFL [19]. We summarize the details of each defense in Appendix A.3.

**Evaluation metrics::** Following previous works [10, 12, 11, 8], we use accuracy & backdoor accuracy (ACC & BAC), attack success rate (ASR), and lifespan to comprehensively evaluate A3FL.

- **ACC & BAC:** We define ACC as the accuracy of the benign global model on clean testing inputs without any attacks, and BAC as the accuracy of the backdoored global model on clean testing inputs when the attacker compromises a part of the clients to attack the global model. Given the dynamic nature of the global model, we report the mean value of ACC and BAC. BAC close to ACC means that the evaluated attack causes little or no impact on the global model utility. A smaller gap between ACC and BAC indicates that the evaluated attack has higher utility stealthiness.

- **ASR:** We embed a backdoor trigger to each input in the testing set. ASR is the fraction of trigger-embedded testing inputs that are successfully misclassified as the target class $\tilde{y}$ by the global model. In particular, the global model is dynamic in FL, resulting in an unstable ASR. Therefore, we use the average value of ASR over the last 10 communication rounds in the attack windows to demonstrate the attack performances. A high ASR indicates that the attack is effective.

- **Lifespan:** The lifespan of a backdoor is defined as the period during which the backdoor keeps effective. The lifespan of a backdoor starts at the end of the attack window and ends when the ASR decreases to less than a chosen threshold. Following previous works [12], we set the threshold as 50%. A long lifespan demonstrates that the backdoor is durable, which means the backdoor remains effective in the global model long after the attack ends. When we evaluate the lifespan of attacks, we extend the FL training process to 3,000 communication rounds.

## 4.2 Experimental Results

**A3FL preserves the utility of the global model:** To verify whether A3FL impacts the utility of global models, we compared their ACCs to BACs. The experimental results on CIFAR-10 are shown in Table 1, where NC denotes Norm Clipping, DSight represents Deepsight, and SFed represents SparseFed. Observe that the maximum degradation in accuracy of the global model caused by A3FL is only 0.28%. Therefore, we can conclude that A3FL preserves the utility of the global model during the attack, indicating that our approach is stealthy and difficult to detect. Similar results were observed in the experiments on TinyImagenet, which can be found in Appendix B.1.

**A3FL achieves higher ASRs:** The attack performances of A3FL and baselines on defenses designed for FL backdoors are presented in Figure 2. The experimental results demonstrate that A3FL achieves higher attack success rates (ASRs) than other baselines. For example, when the defense is Norm

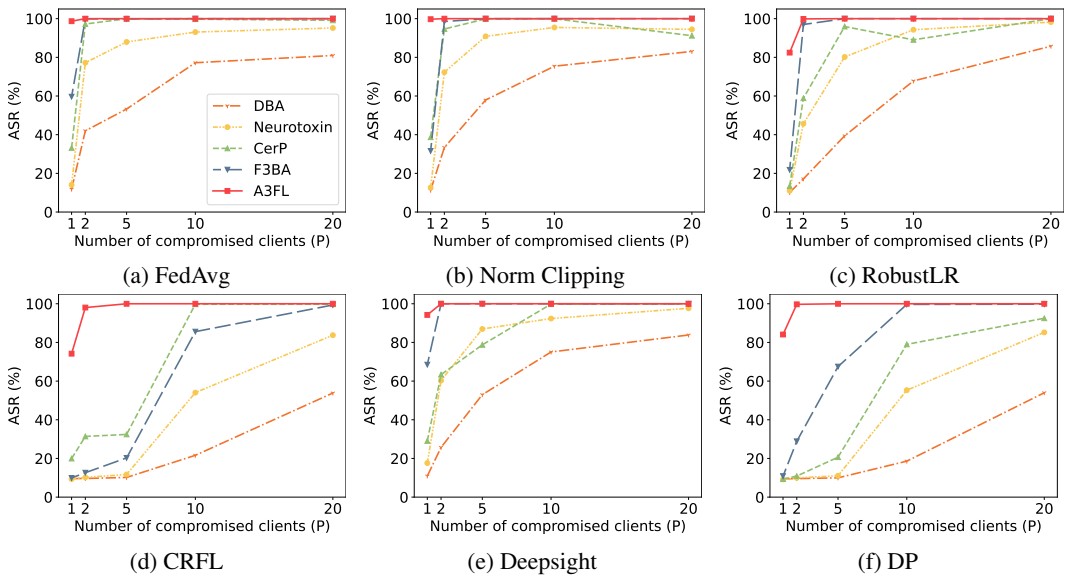

Figure 2: Comparing performances of different attacks on CIFAR-10.

Clipping and only one client is compromised, A3FL achieves an ASR of 99.75%, while other baselines can only achieve a maximum ASR of 13.9%. Other attack baselines achieve a comparable ASR to A3FL only when the number of compromised clients significantly increases. For instance, when the defense is CRFL, F3BA cannot achieve a comparable ASR to A3FL until 10 clients are compromised. We have similar observations on other defenses and datasets, which can be found in Figure 8 and 9 in Appendix B.2.

We note that CRFL assigns a certified radio to each sample and makes sure that samples inside the certified radio would have the same prediction. This is achieved by first clipping the updates $\Delta_t^i$ and then adding Gaussian noise $z \sim \mathcal{N}(0, \sigma^2 I)$ to $\Delta_t^i$. During the inference stage, CRFL adopts majority voting to achieve certified robustness. The strength of CRFL is controlled by the value of $\sigma$. We discuss the performance of CRFL under different values of $\sigma$ in Appendix B.5.

**A3FL has a longer lifespan:** We evaluate the durability of attacks by comparing their lifespans. Recall that the attack starts at the 1,900th communication round and ends at the 2,000th communication round. Figure 3 shows the attack success rate against communication rounds when the defense is Norm Clipping and 5 clients are compromised. As we can observe, A3FL has a significantly longer lifespan than other baseline attacks. A3FL still has an ASR of more than 80% at the end, indicating a lifespan of over 1,000 rounds. In contrast, the ASR of all other baseline attacks drops below 50% quickly. We show more results on other defenses in Appendix B.3 and a similar phenomenon is observed. These experimental results suggest that A3FL is more durable than other baseline attacks, and challenge the consensus that backdoors in FL quickly vanish after the attack ends.

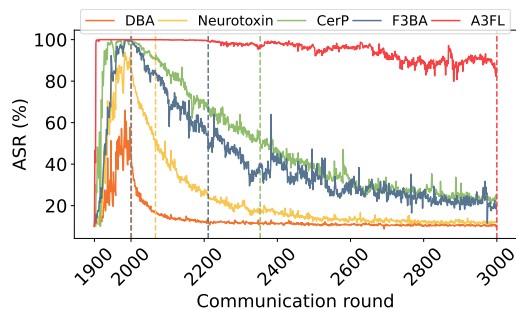

Figure 3: A3FL has a longer lifespan. The vertical dotted lines denote the end of the lifespans of each attack when the ASR of the backdoor drops below 50%. The dotted line at the 100th communication round denotes the end of all attacks.

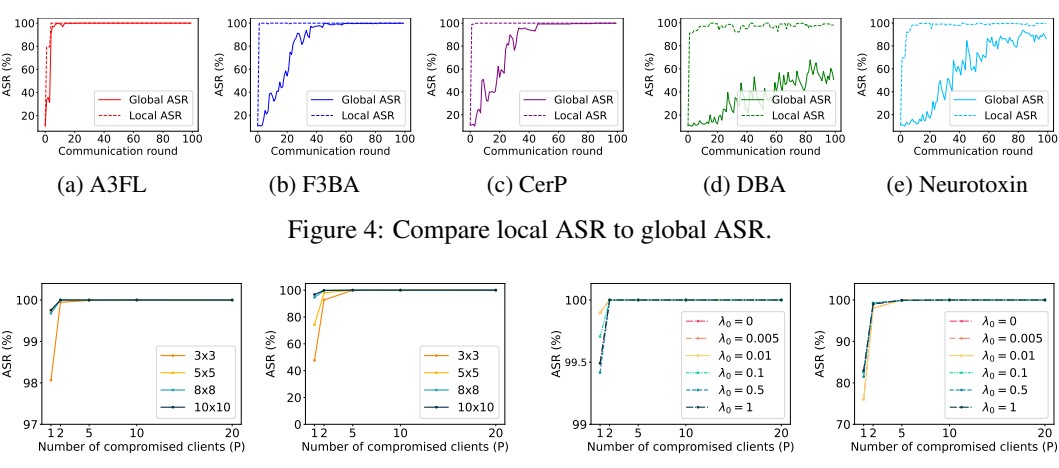

Figure 4: Compare local ASR to global ASR.

Figure 5: The impact of trigger size on the attack performance.

Figure 6: The impact of $\lambda$ on the attack performance.

### 4.3 Analysis and Ablation Study

**A3FL achieves higher ASR when transferred to the global model:** As discussed in Section 3, A3FL achieves higher attack performance by optimizing the trigger and making the backdoor persistent within the dynamic global model. To verify our intuition, we conducted empirical experiments in which we recorded the Attack Success Rate (ASR) on the local model (local ASR) and the ASR on the global model after aggregation (global ASR). For the experiments, we used FedAvg as the default defense and included five compromised clients among all clients.

The results presented in Figure 4 demonstrate that A3FL can maintain a higher ASR when transferred to the global model. While all attacks can achieve high ASR ($\approx 100\%$) locally, only A3FL can also achieve high ASR on the global model after the server aggregates clients' updates, which is supported by the tiny gap between the solid line (global ASR) and the dotted line (local ASR). In contrast, other attacks cannot achieve similarly high ASR on the global model as on local models. For instance, F3BA immediately achieves a local ASR of 100% once the attack starts. But it can only achieve less than 20% ASR on the global model in the first few communication rounds. F3BA also takes a longer time to achieve 100% ASR on the global model compared to A3FL. This observation holds for other baseline attacks. We further provide a case study in Appendix B.8 to understand why A3FL outperforms baseline attacks. In the case study, we observe that 1) A3FL has better attack performance than other baseline attacks with comparable attack budget; 2) clients compromised by A3FL are similarly stealthy to other trigger-optimization attacks. Overall, our experimental results indicate that A3FL is a more effective and persistent attack compared to baseline attacks, which makes it particularly challenging to defend against.

**The impact of trigger size:** We evaluate the performance of A3FL with a trigger size of $3\times3$, $5\times5$, $8\times8$, $10\times10$ respectively (the default value is $5\times5$). Figure 5 shows the impact of trigger size on A3FL. In general, the attack success rate (ASR) improves as the trigger size grows larger. When the defense mechanism is Norm Clipping, we observe that the difference between the best and worst ASR is only 1.75%. We also observe a larger difference with stronger defenses like CRFL. Additionally, we find that when there are at least 5 compromised clients among all clients, the impact of trigger size on the attack success rate becomes unnoticeable. Therefore, we can conclude that smaller trigger sizes may limit the performance of A3FL only when the defense is strong enough and the number of compromised clients is small. Otherwise, varying trigger sizes will not significantly affect the performance of A3FL.

**The impact of $\lambda$:** Recall that $\lambda = \lambda_0 \text{sim}(\boldsymbol{\theta}_t', \boldsymbol{\theta}_t)$. We varied the $\lambda_0$ hyperparameter over a wide range of values to learn the impact of the balancing coefficient on attack performance and record results in Figure 6. Observe that different $\lambda_0$ only slightly impact attack performances with fewer compromised clients. When there are more than 5 compromised clients, the impact of $\lambda_0$ is unnoticeable. For

instance, when the defense is Norm Clipping, the gap between the highest ASR and the lowest ASR is merely 0.5%. We can thus conclude that A3FL is insensitive to variations in hyperparameter $\lambda_0$. We further provide an ablation study in Appendix B.4 for more analysis when the adversarial adaptation loss is disabled, i.e., $\lambda_0 = 0$.

## 5 Conclusion and Future Work

In this paper, we propose A3FL, an effective and durable backdoor attack to Federated Learning. A3FL adopts *adversarial adaption loss* to make the injected backdoor persistent in global training dynamics. Our comprehensive experiments demonstrate that A3FL significantly outperforms existing backdoor attacks under different settings. Interesting future directions include: 1) how to build backdoor attacks towards other types of FL, such as vertical FL; 2) how to build better defenses to protect FL from A3FL.

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

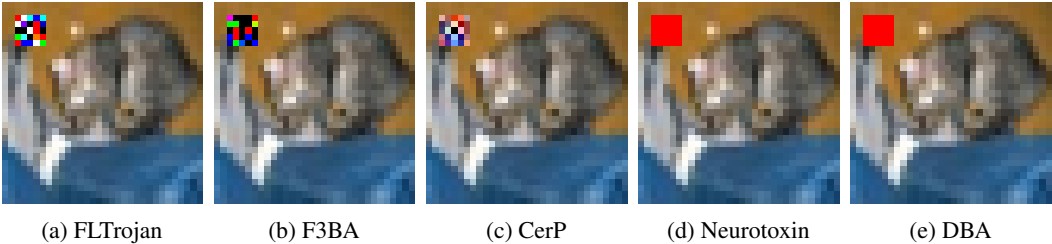

| (a) FLTrojan | (b) F3BA | (c) CerP | (d) Neurotoxin | (e) DBA |

Figure 7: Trigger patterns of evaluated attacks on FedAvg, with P = 2 compromised clients.

## A  Additional Experiment Details

### A.1  Experimental Setup in Figure 1

The preliminary experiment in Figure 1 has the same experimental setup as described in Section 4.1. In particular, We use FedAvg [2] as the server-side aggregation rule. We set the number of compromised clients $P = 1$ in the preliminary experiment. We denote the attack success rate on the global model as global ASR. We further denote the ASR on the local model after local training as the local ASR. When the compromised client is selected by the server, we calculate and update the local ASR after the compromised client optimizes the backdoor trigger and trains its local model on the poisoned local training dataset.

### A.2  Details of Attacks

**A3FL:**  A3FL formulates the trigger optimization as a bi-level optimization problem. A3FL jointly optimizes the adversarial model $f_{\theta'_t}$ with the trigger pattern $\Delta$. A3FL optimizes the adversarial model using SGD with a learning rate of 0.01, a momentum of 0.9, and a weight decay of 0.0005. A3FL updates the trigger pattern using PGD with a step size of 0.01 until convergence. We show the trigger pattern of A3FL in Figure 7a.

**F3BA [10]:**  F3BA directly manipulates a part of local model weights to inject the backdoor via sign flipping. F3BA further jointly optimizes the trigger pattern and the local model weights to maximize the difference between latent representations of clean and backdoored samples, thus achieving higher attack performance. The trigger of F3BA is a squared patch. We show the trigger pattern of F3BA in Figure 7b.

**CerP [9]:**  CerP jointly optimizes the trigger pattern and the local model weights to improve the backdoor effectiveness. Furthermore, CerP aims to improve the backdoor stealthiness by adopting L2-norm regularization to limit the difference between local model weights and global model weights. Therefore CerP can tune the local model to fit the backdoor-poisoned data without inducing large biases in the local model weights. The trigger of CerP is shown in Figure 7c.

**Neurotoxin [12]:**  Neurotoxin only updates unimportant model weights to avoid conflicts with other clean clients. The importance of model weights is determined by the magnitude of their gradients. Model weights with a higher gradient in previous rounds are considered to be more important (frequently updated by other clients). Following the settings in [12], we only update the last 95% important model weights. Neurotoxin uses a fixed trigger pattern, as shown in Figure 7d.

**DBA [11]:**  DBA is a distributed backdoor attack designed to utilize the distributed nature of FL. DBA splits the trigger into different clients. Each client uses a different trigger to attack the FL system during the training stage. In the inference stage, the attacker uses the joint trigger to activate the injected backdoor. The trigger in [11] was designed as several parallel white lines placed at the upper left corner of the input images. This trigger design is not compatible with our attack setting and we can hardly control the attack budget introduced by the trigger following [11]. Therefore in our implementation, we also use a squared patch as the trigger for DBA, as shown in Figure 7e. We randomly split the squared patch into four sub-triggers and these sub-triggers are iteratively used during the attack.

## A.3 FL defenses

**Norm Clipping (NC) [17]:** NC clips clients' updates that are larger than a pre-defined threshold. NC can effectively limit clients' behavior to prevent the global model from being overwhelmed by a few clients. By default, we set the threshold to 1.

**(weak) Differential Privacy (DP) [17]:** DP adds Gaussian noise $z \sim \mathcal{N}(0, \sigma^2 I)$ to clients' updates to perturb carefully crafted malicious updates. Note that this defense is not designed for privacy, so the Gaussian noise is relatively smaller than that adopted in differential privacy. By default, we set $\sigma = 0.002$.

**Robust Learning Rate (RLR) [18]:** RLR aims to maximize the agreement on updating direction across clients to mitigate potential attacks. It is inspired by that the behavior of a compromised client is commonly different from other benign clients. For instance, a compromised client may want to enlarge some model parameters while most benign clients are trying to reduce them. When clients disagree on the updating direction of a parameter, RLR flips the learning rate on the parameter to maximize the loss instead.

**CRFL [19]:** CRFL adopts three techniques to mitigate backdoor attacks on FL. CRFL first clips clients' updates as Norm Clipping does. In our experiments, we set the clipping threshold as 1. CRFL then adds Gaussian noise $z \sim \mathcal{N}(0, \sigma^2 I)$ to clients' updates as DP does. In our experiments, we set $\delta = 0.002$ and we discuss the impact of $\sigma$ on CRFL in Appendix B.5. Finally, CRFL creates several perturbed models by adding independently sampled Gaussian noise to the global model and adopts majority voting for prediction. In our experiments, CRFL creates 5 different perturbed models for prediction at each FL communication round.

**Median [22]:** Median uses the coordinate-wise median value of updates from all clients to update the global model. Median can effectively exclude clients that upload overwhelming updates. However, the Median tends to heavily degrade the model utility.

**Deepsight [20]:** Deepsight adopts three different distance matrices to measure the distances between each client. Deepsight then clusters clients according to different distance matrices and only accepts clients that are in the same cluster across different matrices. The first distance matrix is smaller when the updates in the last layer from clients are similar. The second distance matrix is the L2 distance between the last layer's weight across each client. The third distance matrix is the L2 distance between the outputs of two local models given a batch of randomly generated input images. Deepsight adopts DBSCAN [48] to cluster selected clients. Finally, clusters including potentially malicious clients that have a larger distance from other clusters will be excluded. In our experiments, we set the batch size of randomly generated inputs to 256.

**Bulyan [23]:** Bulyan first excludes potentially malicious clients from all selected clients and then uses the coordinate-wise median value of updates from remaining clients to update the global model. In the first step, $2f$ clients with the highest pairwise Euclidean distances are excluded. In the second step, Bulyan picks $M - 4f$ clients from the remaining $M - 2f$ clients that are closest to the median by coordinate. In our experiments, we set $f = 2$.

**FedDF [24]:** FedDF uses the mean output of all client models as the supervisory signal to distill the next round global model. In particular, FedDF firstly aggregates all selected clients (the same as FedAvg) to obtain a teacher model. Then the server trains the global model to minimize the Kullback Leibler divergence between the logits of the global and teacher model on a set of unlabeled inputs. In our experiments, the learning rate for updating the global model is 0.002 and we train the global model for one epoch at each FL communication round.

**FedRAD [25]:** FedRAD is an extension of FedDF, which assigns a weight to each client model based on their median scores. These scores indicate the frequency with which the prediction of the client model becomes the median value of predictions from all client models. FedRAD then utilizes weighted model aggregation to produce the next round global model. In our experiments, we also update the global model with a learning rate of 0.002 for one epoch at each FL communication round.

**Krum [21]:** Krum selects clients that have the smallest L2 distances to other clients. Only the clients selected by Krum will be used to update the global model. Since Kurm drops most updates from clients, it can achieve strong robustness. However, Krum also affects the accuracy of the model.

Table 2: A3FL maintains the utility of global models on TinyImageNet.

| Defense | FedAvg | NC | RLR | Median | DSight | Bulyan | Krum | SFed | CRFL | DP | FedDF | FedRAD |
|---|---|---|---|---|---|---|---|---|---|---|---|---|
| ACC(%) | 55.45 | 55.31 | 55.34 | 17.12 | 53.71 | 11.19 | 42.87 | 57.39 | 53.58 | 53.38 | 25.31 | 23.12 |
| BAC(%) | 55.25 | 54.98 | 55.28 | 20.92 | 53.44 | 7.33 | 42.35 | 57.08 | 53.45 | 53.17 | 24.90 | 22.57 |

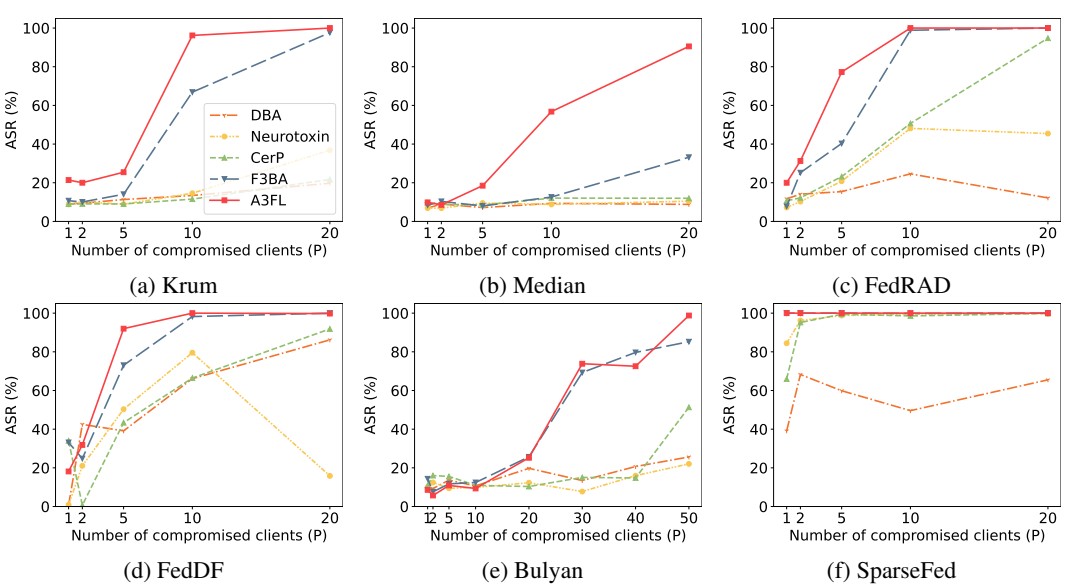

(a) Krum    (b) Median    (c) FedRAD

(d) FedDF    (e) Bulyan    (f) SparseFed

Figure 8: Comparing performances of different attacks on CIFAR-10.

**SparseFed [26]:** SparseFed is proposed to mitigate model poisoning attacks in FL. SparseFed aggregates client updates normally but only updates the top-k highest magnitude elements. It is inspired by that attackers commonly move in distinct directions from the majority of clean clients. Therefore the top-k highest magnitude elements involve less poisoned updates from attackers. In our experiments, we update the top-95% highest magnitude elements.

**FLAME [27]:** FLAME adopts dynamic clustering, adaptive clipping, and adaptive noising to exclude potentially malicious clients. Following the settings for image classification [27], we set $\epsilon = 3705$ and $\delta = 0.001$ controlling the strength of adaptive noising.

## B   Additional Experimental Results

### B.1   A3FL maintains the model utility

We show the accuracy of the global model on TinyImagenet when the attacker presents (BAD) or not (ACC) in Table 2. In particular, we record the accuracy on clean tasks when no attackers are involved to obtain the accuracy (ACC). We further record the accuracy on clean tasks when there are 20 compromised clients among all clients to obtain the backdoor accuracy (BAC). We set the number of compromised clients $P$ to 20 since more compromised clients are likely to result in a higher decrease in clean accuracy. Therefore if A3FL can maintain the model utility even with 20 compromised clients, we can conclude that A3FL is highly stealthy. Note that we use the mean value of ACC and BAC in the attack window (between the 1,900th communication round and the 2,000th communication round) to verify the utility of global models since the server continuously updates the global model. Therefore, using the mean accuracy as the measurement standard can accurately reflect the impact of attacks on the model utility, and eliminate randomness.

As shown in Table 2, the accuracy of the global model does not degrade much when attackers are presented. This indicates that A3FL preserves the accuracy of global models so it is stealthy enough to not be discovered. The differences between ACCs and BACs are within 0.5% in most cases. The highest drop in clean accuracy is observed when the defense mechanism is Bulyan. However, Bulyan significantly degrades the model's accuracy to only 11.19%. The low accuracy indicates that the

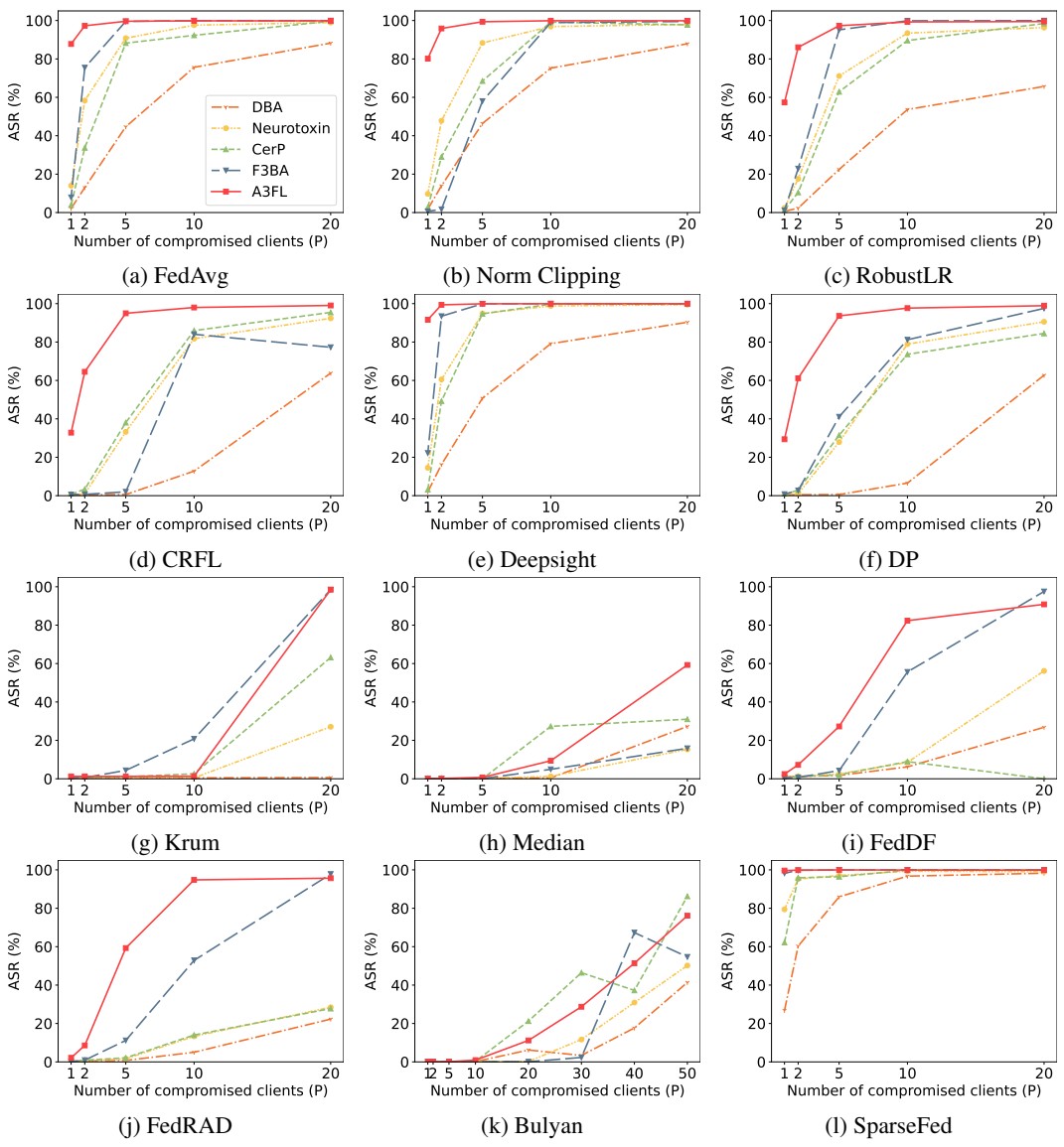

Figure 9: Comparing performances of different attacks on TinyImageNet.

model is highly random, so even though A3FL causes the model's accuracy to drop to 7.33%, we cannot solely conclude that A3FL will reduce the model utility. In general, A3FL does not influence the global model utility. We also observe a similar phenomenon on CIFAR-10, as shown in Table 1.

## B.2   A3FL achieves higher ASRs

We compare the performance of attacks on CIFAR-10 against defenses that are not designed for backdoor attacks in Figure 8. Observe that A3FL achieves the highest ASR under most settings. When the defense is Median, A3FL is the only attack that can achieve high ASR (over 80%). We further show the attacker performance of A3FL on TinyImagenet in Figure 9 and we can observe a similar phenomenon.

## B.3   A3FL has a longer lifespan

In Figure 10, we show that A3FL has a significantly longer lifespan than other baselines with different defenses applied. For instance, when the defense is RobustLR, A3FL can still achieve an ASR of

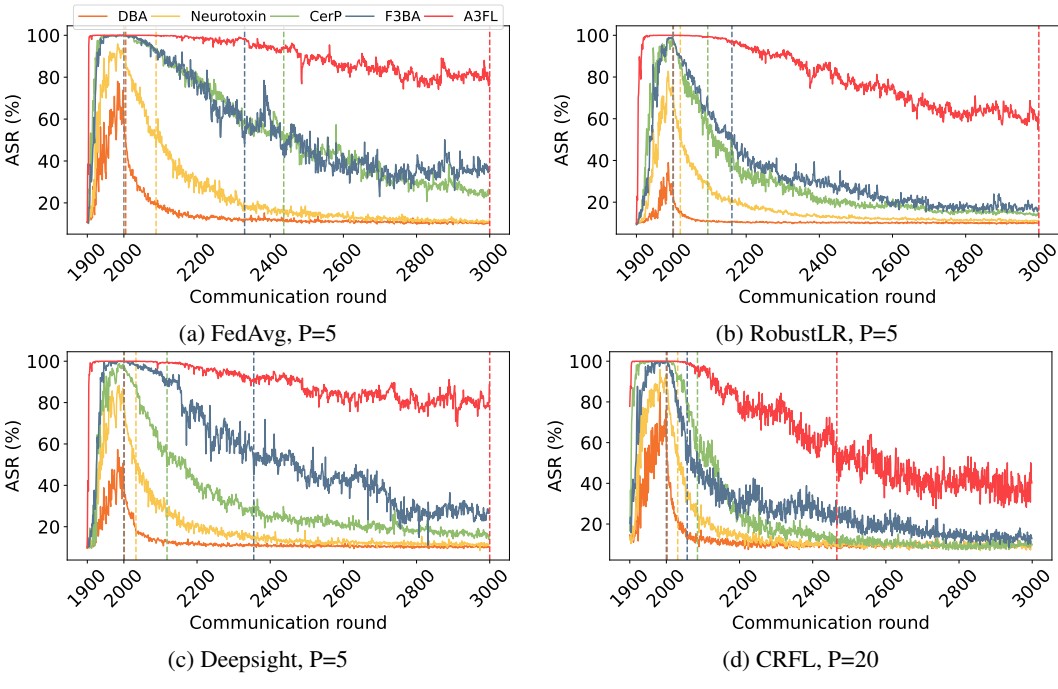

Figure 10: A3FL has a longer lifespan.

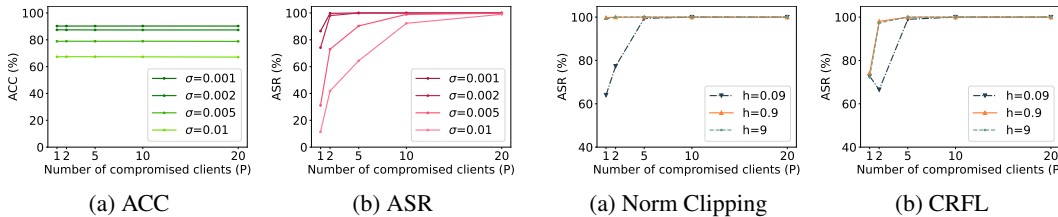

Figure 11: Attack performances against CRFL with different $\sigma$.

Figure 12: Attack performances under different Dirichlet concentration parameters.

62.37% at 1000 rounds after the attack ends. In contrast, the attack success rates of other attacks drop below 50% in less than 150 rounds. Note that when we use CRFL, we set the number of compromised clients $P = 20$ since when there are only 5 compromised clients, all attacks except A3FL failed to achieve high ASR (see Figure 2).

## B.4 Ablation study on component importance

We study the effectiveness of A3FL with or without the adversarial adaptation loss to test the effectiveness of components under FedAvg with $P = 20$ compromised clients among all clients. As shown in Table 3, the adversarial adaptation loss can effectively improve the durability of A3FL. Observe that A3FL can achieve an ASR of 97.66% at 500 communication rounds after the attack and 86.65% at 1,000 communication rounds after the attack. In comparison, A3FL without the adversarial adaptation loss exhibits ASRs that are 4.31% and 15.64% lower than A3FL at these two points.

Table 3: Effect of different components in A3FL.

| ASR(%) ↓ Rounds after attack → | 0 | 500 | 1000 |
|---|---|---|---|
| A3FL without adversarial adaptation | 100.0 | 93.35 | 69.01 |
| A3FL | 100.0 | 97.66 | 84.65 |

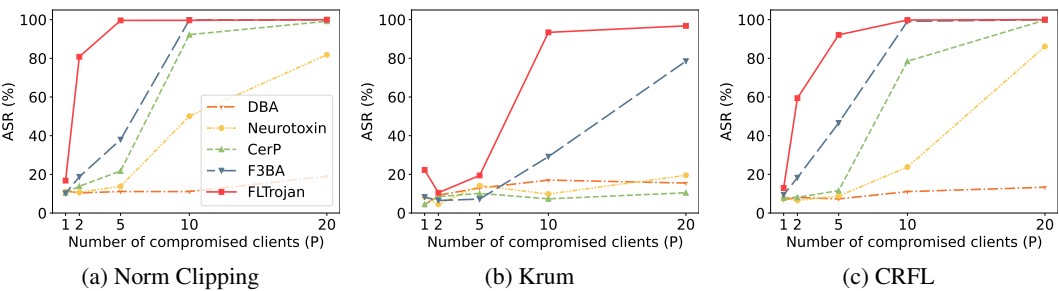

(a) Norm Clipping        (b) Krum        (c) CRFL

Figure 13: Attack performances when the attack starts at the first communication round.

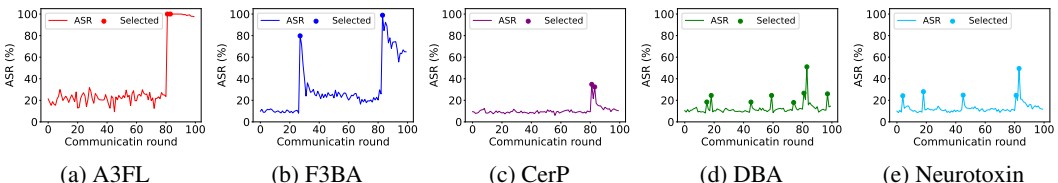

(a) A3FL     (b) F3BA     (c) CerP     (d) DBA     (e) Neurotoxin

Figure 14: ASRs against Krum.

## B.5 Impact of $\sigma$ on CRFL Effectiveness

Figure 11 shows the ACC and ASR when applying CRFL with different $\sigma$. Observe that as the $\sigma$ increases, CRFL can achieve better robustness, indicated by lower ASR. However, the ACC of the global model also drops from 90.25% to 67.33% rapidly, as $\sigma$ increases from 0.001 to 0.01, which is unacceptable. Furthermore, when there are more compromised clients, A3FL can still achieve high ASR even with a large $\sigma = 0.01$. We can thus conclude that CRFL can not sufficiently mitigate A3FL with different $\sigma$.

## B.6 Impact of Data Heterogeneity

We adjust the Dirichlet concentration parameter $h = 0.09, 0.9, 9$ to study whether data heterogeneity influences the performance of A3FL. As shown in Figure 12, A3FL can achieve high ASR regardless of different $h$. When the defense is Norm Clipping and $h = 0.09$, A3FL achieves lower ASR. This can be explained by that a smaller $h$ indicates a more non-i.i.d data distribution. Therefore, the local training set held by the attacker is far from the global data distribution, which increases the difficulty of injecting the backdoor. However, the attack success rate is still high (over 60%) and quickly increases as the number of compromised clients increases.

## B.7 The impact of attack window

We evaluate A3FL against baseline attacks when the attack window starts at the first communication round and ends at the 100th communication round. As shown in Figure 13, A3FL can still remarkably outperform other baseline attacks. For instance, when the defense mechanism is Norm Clipping and there are 5 compromised clients, the gaps of ASR between A3FL and other baseline attacks are at least 62.4%, which is even larger than the gap under default settings. However, we also observe that when the attack starts from the first communication round and there are only a few compromised clients (1 or 2), ASRs of all attacks decrease in comparison to ASRs under default settings. This can be explained by that at the beginning of the training process, the global model changes a lot so the backdoor is easily erased when there are only a few compromised clients. We also observe that when the attack starts from scratch, all attacks fail to have a satisfying lifespan since the model is far from convergence at the 100-th communication round. Therefore, our evaluation of lifespan is conducted following the configuration provided by Neurotoxin [12].

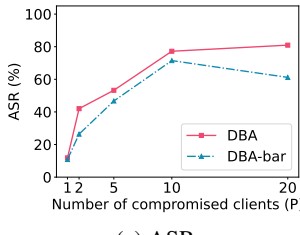 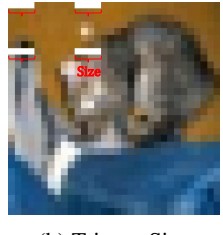 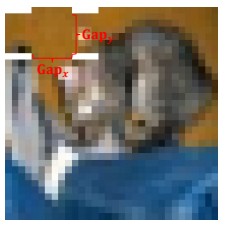 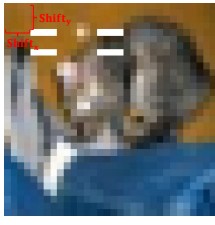

|              |              |              |              |
|:---:|:---:|:---:|:---:|
| (a) ASR | (b) Trigger Size | (c) Trigger Gap | (d) Trigger Location |

Figure 15: Attack performances of DBA using original trigger design. (a) DBA-bar denotes DBA attack with the original trigger design proposed in [11], in which the trigger consists of four white bars. While DBA denotes the DBA attack with the trigger designed as a red square. (b) Trigger size refers to the length of each white bar. (c) Trigger gap $\{\text{Gap}_x, \text{Gap}_y\}$ refers to the distance between each bar. (d) Trigger location $\{\text{Shift}_x, \text{Shift}_y\}$ represents the distance from the trigger to the edge of the image.

## B.8 Case study on Krum

We perform a case study on Krum to gain insight into why A3FL outperforms other baselines. In Figure 14 we record the ASRs and put a "·" notation on the line if Krum selects an attacker-compromised client at that round. Recall that Krum selects one client at each round and only uses the selected client updates to update the global model. Therefore, the chance that a compromised client is selected by the server increases if the backdoor is more stealthy. We have the following observations: 1) fixed-trigger attacks are more frequently selected by the server, while trigger-optimization attacks are selected twice only; 2) fixed-trigger attacks achieve lower ASR even if selected by the server. However, observe that once selected, A3FL quickly achieve 100% ASR, which is because A3FL can maintain higher ASR when transferred to the global model as stated above. A3FL is also durable after being selected, leading to a higher ASR at the end of the attack. In comparison, F3BA is selected on the 26th round and achieves $\approx$ 80% ASR. But the ASR quickly drops after that. CerP is also selected twice, but it cannot achieve as high ASR as A3FL and F3BA do, which is caused by the strict regularization on the local model bias. In addition, the ASR of CerP also drops quickly when the compromised clients are not selected by the server.

## B.9 The impact of DBA trigger pattern

In our experiments, we set the trigger pattern of DBA to be a red square at the upper left corner. However, in [11], the trigger is designed as four white lines. We, therefore, discuss the performance of DBA when using the original trigger design. The original trigger design of DBA is determined by three hyperparameters: trigger size (TS), trigger gap (TG), and trigger location (TL). In particular, the trigger gap consists of a horizontal gap ($\text{Gap}_x$) and a vertical gap ($\text{Gap}_y$). The trigger location consists of a horizontal shift ($\text{Shift}_x$) and a vertical shift ($\text{Shift}_y$). We explain these hyperparameters in Figure 15b,15c, and 15d respectively. Following the default settings in [11], we set $\{\text{TS}, \text{TG}, \text{TL}\} = \{4, (6, 6), (0, 0)\}$.

We compare the attack performance of DBA and DBA-bar (DBA with original trigger design) in Figure 15a. Observe that with the original trigger design, DBA-bar achieves an even lower ASR. This phenomenon supports that the default trigger design in our experiments does not degrade the attack performance of DBA. In contrast, DBA can even achieve a higher ASR without the original trigger design.

## B.10 Transferability of A3FL under different settings

We further evaluate the attack performance of A3FL on more datasets, defenses, and model architectures. In particular, we record the performance of A3FL on FLAME [27] in Figure 16a. We also evaluate A3FL on other model architectures [49] and datasets [43] in Figure 16b,16c. Observe that A3FL can always outperform baseline attacks under different settings.

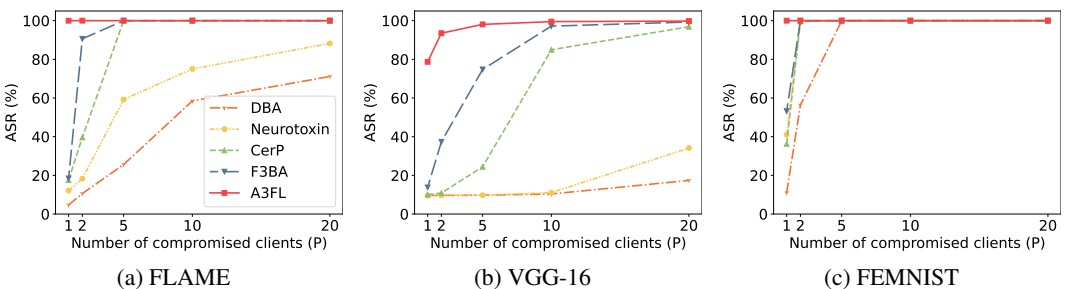

(a) FLAME      (b) VGG-16      (c) FEMNIST

Figure 16: Attack performances of A3FL under different settings.

## B.11    Discussion on Ethical Implications

We admit that the discovery of a new backdoor attack in federated learning results in potential ethical implications. A3FL mainly focuses on image classification, which can be deployed in security-sensitive applications such as human face recognition. However, discovering a new backdoor attack and mitigating its threats is necessary to improve the robustness of the federated learning paradigm. Safeguarding the integrity and ethical dimensions of federated learning is crucial to ensure the best interests of individuals and society. We believe that future work can eliminate the threat of proposed attacks, and it is important to focus research on FL defenses.

