# OpenReview forum: "A3FL: Adversarially Adaptive Backdoor Attacks to Federated Learning"
_NeurIPS.cc/2023/Conference — NeurIPS 2023 poster_

### Official Review · Reviewer_k1k8 · 2023-06-20

**Soundness:** 2 fair
**Presentation:** 2 fair
**Contribution:** 2 fair
**Rating:** 4
**Confidence:** 5

**Summary:**

The paper introduces A3FL, a backdoor attack tailored for FL. Unlike traditional approaches, A3FL adapts backdoor triggers in an adversarial manner considering global training dynamics, thereby enhancing its robustness and effectiveness. By accounting for discrepancies between global and local models in FL, A3FL ensures that the trigger remains potent even if the global model attempts to negate it. The paper demonstrates through extensive experiments that A3FL significantly surpasses existing backdoor attacks in terms of efficiency, even when faced with established defenses.

**Strengths:**

The paper is easy to follow. It addresses a crucial problem in federated learning. The authors have made code available, which makes it easy for others to replicate the study.

**Weaknesses:**

First of all, the paper needs editorial revision as it discusses the same thing multiple times. By making it shorter and to the point, there is more room to include extra results in the main content.

The technical limitations of the paper are discussed as follows:

Line 36-37: "the selected trigger is usually sub-optimal, which makes the attack less effective and stealthy as shown in experiments" - I would like to request some clarification and elaboration. Firstly, it would be beneficial for the readers if the authors could provide references or citations that substantiate the claim regarding the sub-optimality of the triggers in existing backdoor attacks. Additionally, it is crucial to note that in FL, semantic backdoor triggers are considered to be particularly stealthy. The stealthiness of semantic triggers arises because they leverage the inherent properties of images rather than altering pixels, making them less detectable. In this regard, it would be insightful if the authors could elaborate on what they imply by 'stealthy' in the context of semantic triggers.

Line 45-46: "they only leverage local models of compromised clients to optimize the backdoor trigger, which ignores the global training dynamics" - However, it is noteworthy that some existing works, such as Neurotoxin[1], have been known to embed backdoors by considering parameters that remain robust to the training dynamics in FL systems. Please clarify the claims made in this regard and support them with relevant references or citations, particularly contrasting with what Neurotoxin has achieved. Also, differentiate the contribution of A3FL in this regard.

Line 47-50: "they strictly regulate the difference between the local and global model weights to bypass defenses, which in turn limits the backdoor effectiveness" - While regulating the difference between local and global model weights can be a strategy to bypass defenses, it has been demonstrated in the literature that this approach can also lead to high backdoor efficiency[2]. Please provide a more comprehensive analysis of this aspect and substantiate the claim regarding the limitation of backdoor effectiveness in such approaches with references or citations that clearly illustrate this limitation.

Line 170: "injected backdoor is rapidly eliminated since the global model is continuously updated by the server" - This statement is misleading. In reality, the persistence of backdoors in FL systems can be influenced by the timing of their injection. Existing research indicates that if a backdoor is introduced after the model has reached a stable state, it tends to persist longer before being eliminated.

Line 174-175: "the global model is trained to directly unlearn the backdoor" - I would like to request the authors to provide references or citations that substantiate the statement. It is not clear how the backdoor trigger can be eliminated from the global model without knowing the specific patterns that activate it.

In the A3FL workflow, the first step involves optimizing the trigger pattern based on the model parameters. Subsequently, the adversarial global model is optimized in each FL training round. This approach raises a question regarding the trigger pattern during evaluation. Unlike in training rounds where the trigger patterns vary due to FL dynamics, it is unclear what trigger will be used during the evaluation. Specifically, it is important to describe whether the adversary requires access to the global model to generate triggers during testing. If the adversary does require such access, it would be considered an adversarial attack rather than a backdoor attack. Conversely, if the adversary does not need access to the global model, the trigger for misclassification remains uncertain.

A3FL uses an adversarial example generation strategy for the incorporation of backdoor triggers. In contrast, an approach called PerDoor[3] has been recently introduced, utilizing adversarial examples to achieve a similar purpose. It is essential for the authors to clearly explain the contributions and differentiating factors of A3FL in relation to PerDoor.

The efficiency of A3FL must be evaluated in the presence of FLAME[4], a state-of-the-art defense mechanism against backdoor attacks in FL. Furthermore, A3FL needs to be compared against recent state-of-the-art attacks, like 3DFed[5].

A3FL should be evaluated using more practical benchmark federated learning datasets in the LEAF project[7]. The authors have only used the ResNet18 model as the underlying structure for all their experiments. For a more comprehensive evaluation, A3FL should be evaluated across a variety of models.

[1] Z. Zhang et al., "Neurotoxin: Durable Backdoors in Federated Learning", ICML 2022.

[2] H. Wang et al., "Attack of the Tails: Yes,You Really Can Backdoor Federated Learning", NeurIPS 2020.

[3] M. Alam et al., "PerDoor: Persistent Non-Uniform Backdoors in Federated Learning using Adversarial Perturbations", arXiv 2022.

[4] T. Nguyen et al., "FLAME: Taming Backdoors in Federated Learning", Usenix Security 2022.

[5] H. Li et al., "3DFed: Adaptive and Extensible Framework for Covert Backdoor Attack in Federated Learning", IEEE S&P 2023.

[6] https://leaf.cmu.edu/

**Questions:**

Please address the issues discussed in Weaknesses.

**Limitations:**

The paper has several limitations, as discussed in the review.

---

> ### Author Rebuttal · Authors · 2023-08-10
>
> ## reviewer k1k8
> We thank the reviewer for the detailed suggestion to improve the clarity and enhance the evaluation of our work.
>
> > Q1: Line 36-37
>
> Thanks for commenting on the place that could cause misunderstanding. In the context of this sentence (Line 34-37), we were discussing the fixed-trigger backdoor attacks. The sub-optimality of such fixed-trigger attacks has been shown by previous works [1-3] and in Section 4.2 of our paper.
>
> We also thank the reviewer for pointing out the possible confusion w.r.t. the definition of stealthiness. In our case, we consider stealthiness in FL as the capability of bypassing defenses without harming the utility of the global model. We do not care about the stealthiness in images as local data is not directly seen by the server. Following this definition, semantic backdoor trigger is not stealthy since they inherit the sub-optimality of fixed-trigger backdoors. We will clarify the meaning of stealthiness and discuss semantic backdoors in revision.
>
> > Q2: Line 45-46
>
> Thanks for pointing out the place that could cause misunderstanding. In the context of this sentence (lines 44-45), we were discussing the limitations of trigger-optimization backdoor attacks[4][5]. Neurotoxin is a fixed-trigger attack which does not fall into this discussion in Line 45-46.
>
> As discussed in Line 34-37 and 95-96, the differences between A3FL and Neurotoxin are two folds. First, A3FL considers a worst-case scenario in which the global model is trained to directly unlearn the backdoor while Neurotoxin does not have such an adversarial component. A3FL further optimizes the backdoor trigger to survive the worst case scenario, therefore improving the backdoor persistence. In comparison, Neurotoxin passively avoids embedding the backdoor in frequently changing model weights. As shown in Figure 17 of [6], Neurotoxin can only achieve minor improvement in the image domain, which is also supported by our experiments in Figure 2.
>
> > Q3: Line 47-50
>
> We would like to use the ablation study in F3BA[4] to demonstrate our claim. In Appendix D.6 of [4], the authors wrote "Meanwhile, too high candidate parameters proportion for fully-connected layers can cause an obvious loss of ASR." The authors also conducted a grid search to find an optimal regularization proportion. This indicates that regularization can potentially harm the backdoor efficiency. We conduct an experiment to further support our claim. We use L2-norm regularization to regulate the backdoor loss and vary the strength of regularization by adjusting the value of balancing coefficient $\beta$. As observed in Table 7 in the attached PDF, the ASR drops as the strength of regularization enhances. We will include this in the revised paper.
>
> > Q4: Line 170
>
> Thanks for suggesting another factor (the timing of injection) that influences the persisitence of attack. We were focusing on the attack design perspective, when the injection timing is controled the same, other baseline backdoors are eliminated in a shorter time than A3FL. We will modify this sentence into "injected backdoor could be eliminated since the global model is continuously updated by the server, especially when the global model has not reached a stable state".
>
> > Q5: Line 174-175
>
> The FL server cannot access the specific backdoor trigger in practice. However, the compromised client has access to the trigger as an attacker to simulate a worst-case scenario where the global model is trained to directly unlearn the backdoor trigger, as discussed in lines 173-174.
>
> > Q6: The trigger pattern during the evaluation and the attacker's access to global model.
>
> As discussed in Line 248-252, in the attack window (i.e. compromised clients being selected to participate), the attacker can access the global model through compromised clients to optimize the trigger. After the end of the attack window, the attacker-compromised clients leave the FL procedure. The attack performance is evaluated using the trigger at the end of the attack window. Following [6], the trigger pattern will no longer be optimized after the attack window, therefore the attacker do not need to access the global model anymore.
>
> > Q7: Compare A3FL to PerDoor
>
> We will discuss the difference between PerDoor[15] and A3FL in revision. PerDoor adopts adversarial attack to generate the backdoor trigger. However, the trigger in A3FL is not an adversarial example. As discussed in Section 3, we optimize the backdoor trigger to survive adversarially crafted global model. This pipeline can be seen as adversarial training in a reverse way where the crafted global model corresponds to the inner maximization problem and the trigger optimization corresponds to the outer minimization problem.
>
> > Q8: More baselines: FLAME and 3DFed.
>
> We will discuss FLAME[13] in the revised paper. As for evaluation, we found the code of FLAME is not publicly available and have not got response from the author after acquiring the code. We will update the experimental results on FLAME once the code is received.
>
> We will discuss 3DFed[14] in the revised paper. However, we note that 3DFed was first published in S&P 2023 held on 21-15 May 2023. And the paper was added to IEEE on 21 July 2023. To the best of our knowledge, 3DFed was not published on arxiv before. Therefore, we were not able to compare A3FL against 3DFed. What's more, according to the concurrent policy of NeurIPS 2023, we are not required to consider concurrent work which was published in two months before the submission deadline. We also found that the code of 3DFed is not publicly available and will update the experimental results once the code is received.
>
> > Q9: A3FL should be evaluated using more practical benchmark in the LEAF project.
>
> Thanks for suggesting more benchmarks. In the attached PDF, we further report the performance of A3FL on FEMNIST from the LEAF project in Table 3, as well as A3FL on VGG16 in Table 2. Observe that A3FL can still achieve high ASRs.

---

> > ### Author Response · Authors · 2023-08-17
> >
> > >Supplementary results to respond **Q8: More baselines: FLAME and 3DFed**.
> >
> > We would like to follow up on this question and welcome further discussion from the reviewer. Since there is no official implementation of FLAME [13] available yet, we tried our best to reproduce FLAME following the paper and evaluate A3FL against FLAME. Following the settings for image classification in Appendix B.3 of [13], we set $\epsilon = 3705$ and $\delta = 0.001$. We adopt `sklearn.cluster.HDBSCAN` as the implementation of the clustering algorithm adopted by FLAME. We set `min_cluster_size = N/2+1` and `min_samples = 1` following Appendix.E in [13], where N is the number of sampled clients in each round. The experimental results are shown below (and recall P is the number of compromised clients):
> >
> > | P          | 1     | 2     | 5     | 10    | 20    |
> > |------------|-------|-------|-------|-------|-------|
> > | Neurotoxin | 9.74  | 10.34 | 37.24 | 92.41 | 97.7  |
> > | DBA        | 9.46  | 11.22 | 13.33 | 55.96 | 90.61 |
> > | CerP       | 18.55 | 22.41 | 88.75 | 99.76 | 99.85 |
> > | F3BA       | 10.76 | 11.55 | 63.63 | 99.99 | 99.98 |
> > | A3FL       | **61.75** | **91.97** | **100**   | **100**   | **100**   |
> >
> > Observe that A3FL can still achieve the highest ASR against FLAME. We will include the evaluation in the revised paper.

---

> ### Comment · Reviewer_k1k8 · 2023-08-21
>
> I want to thank the authors for their efforts in responding to these queries. I respect the hard work put into this paper and trust that these suggestions will only enhance its quality. I am satisfied with most of the responses and am increasing my score.
>
> However, I have concerns over a couple of points. (1) I am not satisfied with the discussion that differentiates Neurotoxin from A3FL, considering the worst-case assumption. (2) The discussion on trigger unlearning is still not convincing, considering the threat model.

---

> > ### Author Response · Authors · 2023-08-21
> >
> > We thank the reviewer for the response and we are glad that most of the concerns are addressed. We hope that the following clarification could further alleviate your remaining concerns.
> >
> > > (1) I am not satisfied with the discussion that differentiates Neurotoxin from A3FL, considering the worst-case assumption.
> >
> > We will avoid the imprecise word of "worst-case" (as can be seen in the discussion with AC) yet we believe our distinction with Neurotoxin is clear and is not really relevant to this issue. We will try our best to iterate the differences as follows:
> >
> > Neurotoxin uses a **fixed trigger**, focuses on how to identify parameters that are less important to modify, thus improves the durability of injected backdoor. In comparison, A3FL is a **trigger-optimization backdoor attack** focusing on how to optimize a persistent backdoor trigger that withstands potential defenses. This is achieved by simulating the defender's goal (i.e., mitigating the impact of the backdoor trigger), and optimize the trigger accordingly to survive possible defenses. While we agree that both Neurotoxin and A3FL share the goal of improving the durability of the trigger, the intution as well as the technical details significantly dispart. We will discuss and clarify this in the revised paper.
> >
> > > (2) The discussion on trigger unlearning is still not convincing, considering the threat model.
> >
> > We appreciate the reviewer's comment but it is not very clear on what part is unconvincing to the reviewer. We will try our best to answer this with our understandings.
> >
> > We would like to emphasize the threat model: the real server does not have access to the backdoor trigger and any private data held by clients. Meanwhile, the malicious client does not know the server-side defense strategy or movement. Thus from the malicious client's perspective, it is hard to guess what the server might do. Instead, the malicious client can only simulate the defender's goal: mitigating the impact of backdoors (i.e., through trigger unlearning on the client side). The simulation is feasible because: 1) the malicious client has the actual trigger; 2) the malicious client has the received global model from the server.
> >
> > Again, this is a simulation from the client side (i.e., the malicious client use the server model copy as well as its own data/trigger to mimic what could happen after the defense). The real server does not have access to the trigger and trigger unlearning is done by the malicious clients. We hope this clarifies your concern.

---

### Official Review · Reviewer_BFfe · 2023-07-04

**Soundness:** 3 good
**Presentation:** 3 good
**Contribution:** 2 fair
**Rating:** 5
**Confidence:** 4

**Summary:**

The authors introduce A3FL, a backdoor attack that strategically adjusts the backdoor trigger to decrease the chances of its removal by the global training dynamics. The fundamental idea behind this approach lies in the disparity between the global model and the local model in Federated Learning (FL), which diminishes the effectiveness of locally optimized triggers when transferred to the global model. To address this issue, the authors tackle the optimization of the trigger in a manner that ensures its survival even in the worst-case scenario, wherein the global model is specifically trained to eliminate the trigger. Through extensive experiments conducted on benchmark datasets, they comprehensively evaluate the efficacy of A3FL against twelve existing defense mechanisms.

**Strengths:**

1. The authors consider a bi-level objective (i.e., the worst-case adaptation of global dynamics) to optimize the trigger pattern during the FL process without knowing the defense mechanism.
2. Experiment results are good.
3. The proposed method compares with extensive baselines in different settings.

**Weaknesses:**

1. No theoretical analysis or guarantee on attack performance and convergency/sample complexity for Algorithm 1
2. It seems like solving a bi-level optimization in each FL round is computational inefficient. Either empirical or theoretical justifications comparing with baseline methods are needed to prove the efficiency.
3. To incorporate global dynamics (i.e., long-term goal, benign clients' behaviors, defense mechanism, etc.) is useful to increase backdoor durability, however, it has already been proposed in previous works [1] [2].

[1] Wen, Yuxin, et al. "Thinking two moves ahead: Anticipating other users improves backdoor attacks in federated learning." arXiv preprint arXiv:2210.09305 (2022).
[2] Li, Henger, et al. "Learning to backdoor federated learning." arXiv preprint arXiv:2303.03320 (2023).

**Questions:**

1. The calculation of $\theta'_t$ solely based on the local training data of malicious clients appears to be an approximation of the true worst-case scenario, as it neglects the inclusion of benign clients' data. I am interested in understanding the implications of an inaccurate $\theta'_t$ on the performance of the attack, particularly in scenarios where there is a high level of heterogeneity. It would be intriguing to compare the attack performance when the attacker possesses knowledge of the benign workers' data, considering previous studies have demonstrated that attackers can learn and reconstruct benign workers' data through inference attacks [1] [2].
2. How to choose/tune a good or even an optimal (is it exist?) $\lambda$? It seems to me in each FL round, the optimal $\lambda$ should be different (based on current model, number of attackers been chosen, etc.)
3. To ensure fair comparison, while A3FL tune parameters like $\lambda$ and poison ratio, how are the hyperparameters are chosen in other baseline attack methods?

[1] Geiping, Jonas, et al. "Inverting gradients-how easy is it to break privacy in federated learning?." Advances in Neural Information Processing Systems 33 (2020): 16937-16947.
[2] Li, Henger, Xiaolin Sun, and Zizhan Zheng. "Learning to attack federated learning: A model-based reinforcement learning attack framework." Advances in Neural Information Processing Systems 35 (2022): 35007-35020.

**Limitations:**

1. It is unclear to me what exactly is being optimized in equations (2) and (3). I assume that the optimization pertains to the parameters associated with each pixel within the predetermined square pattern. In the conducted DBA experiments, the trigger(s) take the form of fixed-sized square(s) located at specific position(s) and may involve certain numbers of squares. These characteristics represent potential parameters that can be incorporated into the optimization problem. However, it is worth noting that for achieving the most versatile trigger, it is worthwhile to consider every pixel in the image. In recent works [1] [2], researchers have explored generated backdoor triggers that encompass the entire range of the image.
2. Post-training stage defenses play a vital role in countering backdoor attacks. Even within the context of FL, certain techniques such as Neuron Clipping [3] and Pruning [4] have demonstrated their effectiveness in detecting and mitigating the impact of backdoor attacks. Consequently, I am curious to know how the proposed A3FL performs when subjected to these post-training stage defenses.

[1] Salem, Ahmed, et al. "Dynamic backdoor attacks against machine learning models." 2022 IEEE 7th European Symposium on Security and Privacy (EuroS&P). IEEE, 2022.
[2] Doan, Khoa D., Yingjie Lao, and Ping Li. "Marksman backdoor: Backdoor attacks with arbitrary target class." Advances in Neural Information Processing Systems 35 (2022): 38260-38273.
[3] Wang, Hang, et al. "Universal post-training backdoor detection." arXiv preprint arXiv:2205.06900 (2022).
[4] Wu, Chen, et al. "Mitigating backdoor attacks in federated learning." arXiv preprint arXiv:2011.01767 (2020).

---

> ### Author Rebuttal · Authors · 2023-08-10
>
> ## reviewer BFfe
>
> We thank the reviewer for the constructive comments to strengthen our work.
>
> > Q1: No theoretical analysis for Algorithm 1.
>
> While theoretical analysis is important and interesting, it is very challenging and an open problem to theoretically analyze the performance of FL backdoor attacks, especially when A3FL is implemented through bi-level optimization. To the best of our knowledge, most existing works proposing FL backdoor attacks did not provide theoretical analysis on attack performance or convergency. We will leave this as our future work.
>
> > Q2: Solving a bi-level optimization in each FL round is computational inefficient.
>
> Thank the reviewer for constructive comments. We empirically compare the efficiency of A3FL to other baseline attacks. We record the average time taken by each attack in each round in Table 4 in the attached PDF. Observe that A3FL has a comparable efficiency with CerP and F3BA.
>
>
> > Q3: Incorporating global dynamics has already been proposed in previous works.
>
> Thank you for commenting on the place that could cause misunderstanding. We will discuss and comment these works in the revision. Instead of incorporating global training dynamics, A3FL proposes a different principle by anticipating a worst-case scenario in which the injected backdoor is directly unlearned by the global model. A3FL optimizes the backdoor trigger to make it robust to this worst-case scenario and therefore persistent to other perturbations introduced by global training dynamics. Our empirical experiments in Section 4.3 and Appendix B.8 demonstrates this intuition. As observed in Figure 4, while all backdoor attacks can achieve nearly 100% ASR on the local model, only A3FL can achieve similarly high ASR when transferred to the global model. This observation demonstrates that A3FL is more persistent to perturbation when transferred to the global model.
>
> > Q4: Attack performance when the attacker possesses knowledge of the benign workers' data.
>
> Thank you for the interesting perspective. We discussed the impact of data heterogeneity in Appendix B.6. As shown in Figure 12, a large data heterogeneity does not significantly impact the attack performances of A3FL.
>
> We further consider a scenario where the attacker can access a part of private dataset from benign clients. Specifically, we merge the private training dataset of 5 randomly selected benign clients as $\mathcal{D}_p$. We assume that the attacker can access $\mathcal{D}_p$ to better approximate the worst-case scenario. As observed in Table 5 in the attached PDF, A3FL with access to benign worker's data indeed has slightly better performance but the improvement is marginal. We will discuss these works[7][8] and the potential extension in the revision.
>
>
> > Q5: How to tune $\lambda$
>
> As discussed in Line 261-269, we introduce a balancing coefficient $\lambda_0$ to control the strength of adversarial training, and $\lambda=\lambda_0$sim$(\theta'_t, \theta_t)$. When the adversarial global model $\theta'_t$ differs a lot from the current global model $\theta_t$, we pay less attention to the adversarial training loss. This is motivated by that when the adversarial global model too different, the backdoor trigger optimized on current model is harder to be adapted to the adversarial model thus can be more easily unlearned. In practice, we find this helpful in balancing the bi-level optimization. As discussed in Section 4.3, Figure 6, A3FL is not obviously sensitive to different values of $\lambda_0$. We simply grid search $\lambda_0$ on FedAvg and apply the same value of $\lambda_0$ to other cases. Since we observe that this has already achieved satisfying results, we do not adopt more complicated tuning methods to find a better $\lambda_0$.
>
> > Q6: Hyperparameter settings of other attacks.
>
> Thank you for commenting on the place that could cause confusion. For all evaluated attack methods, we tune hyperparameters on FedAvg via grid search. It is reasonable to search for optimal hyperparameters on FedAvg and use the same setting for other experiments, since the attacker does not know the server-side defense beforehand.
>
> We have discussed hyperparameter settings in Appendix A.2 and B.9. For CerP, we set two balancing coefficients $\alpha = 0.005$ and $\beta = 0.005$. For F3BA, we set the proportion of candidate weights for convolutional layers to be 0.02, and the proportion for fully connected layers to be 0.001. For Neurotoxin, we only update the last 95% important model weights. For DBA, we set the trigger shift to be 0, trigger gap to be 6, and trigger size to be 4.
>
> > Q7: What exactly is being optimized in equations (2) and (3).
>
> We thank the reviewer for suggesting an interesting direction to combine the progress from both optimization (as our work does) and trigger pattern design perspective. To clarify, our work mainly focuses on the optimization perspective, thus uses the commonly adopted squared trigger shape: A3FL only optimizes $\delta$ for the masked-out trigger region. This can be viewed as applying $\delta = \delta \cdot m$ at the end of each time of optimization. $m$ is a binary mask with 1 on the pixels within the predefined region. We adopted the same trigger shape for evaluated attacks to ensure fair comparison. We will explain this in the revised paper to eliminate potential confusion. We will also discuss these papers[9-12] in the revised paper, and extend A3FL to backdoor triggers encompassing the entire range of the image in our future work.
>
> > Q8: Post-training stage defenses should be considered.
>
> Thank you for your constructive comment. Following the suggestion, we evaluate A3FL against Neuron Clipping and include the experimental results in Table 6 in the attached PDF. We observe that Neuoron Clipping cannot impact the ASR of A3FL. A3FL still significantly outperforms other baselines against this defense. We will include this defense in the revised paper.

---

> > ### Comment · Reviewer_BFfe · 2023-08-18
> >
> > I appreciate the authors' response and informative clarification. After reading the rebuttal and other reviewers' comments, most of my concerns have been addressed. I will change my rating. Thank you.

---

> > > ### Author Response · Authors · 2023-08-19
> > >
> > > Thanks for your feedback. Your constructive comments and suggestions are exceedingly helpful to improve our paper. Please let us know if you have any further suggestions.

---

### Official Review · Reviewer_i2mP · 2023-07-06

**Soundness:** 4 excellent
**Presentation:** 3 good
**Contribution:** 4 excellent
**Rating:** 7
**Confidence:** 4

**Summary:**

This work proposes a better backdoor attack on Federated Learning. Existing FL backdoor attacks do not consider the global training dynamics, resulting in limited backdoor attack performance. As the true global training dynamics are impossible to know, this work proposes a method to regularize backdoor training using the worst-case global training dynamics as guidance. The worst-case scenario represents a strong protector who tries to unlearn the exact backdoor pattern, and the proposed backdoor attack greatly benefits from adversarially adapting to this worst case. The paper demonstrates its better performance against 12 existing defense approaches and consistently outperforms state-of-the-art (SOTA) backdoor attacks by a large margin (10x), especially when there is only a small number of attackers.

**Strengths:**

1. The method successfully unlearns the backdoor and learns a strong backdoor trigger to adapt (adversarially) to such global learning objectives.
2. The design is smart in automatically tuning the lambda, which uses similarity to adjust. This design considers potential defenses and explains why the proposed method works well against them. Additionally, the tuning of the base lambda is not sensitive, as shown in Fig 6.
3. The proposed method has been shown to be effective against 12 existing defense approaches and consistently outperforms state-of-the-art works by a large margin (10x).
4. There is no need for strict regularization, and it is harmless to utility.
5. It requires a smaller attack budget in terms of the number of attackers and the number of rounds in poisoning.
6. A comprehensive evaluation is done on utility, ASR, and lifespan.
7. The achieved better transferability of the backdoor to the global model is well analyzed.

**Weaknesses:**

1. Different from previous state-of-the-art works, this work does not directly consider potential defenses (i.e., detection-based methods), while still demonstrating better performance against them. The reason for this is not well justified.
2. The impact of the number of available training data on the attack performance is not clear.


**Questions:**

1. Can you further address the weaknesses and limitations?
2. Can you provide some directions regarding potential defense strategies?

**Limitations:**

1. The trigger pattern is limited to a rectangular shape. It would be interesting to explore the adoption of irregular shapes in the proposed attack.
2. More datasets and architectures can be considered.

---

> ### Author Rebuttal · Authors · 2023-08-10
>
> ## reviewer i2mP
> We appreciate the positive comments on our paper and insightful suggestion for further improvements from the reviewer.
>
> > Q1: Different from previous state-of-the-art works, this work does not directly consider potential defenses (i.e., detection-based methods), while still demonstrating better performance against them. The reason for this is not well justified.
>
> We are sorry for the confusion. We provide two reasons in terms of method design and empirical observation.
>
> In the method design, while we do not explicitly consider any specific defense, we design the adversarial adaptation loss in Eq. (3) to enables the attacker to foresee and survive even the worst-case scenario where the global model is trained to directly unlearn the backdoor (which is imaginary since in practice the server does not know the backdoor trigger, and thus it is considered to be stronger than any existing defense). Therefore, if the attacker could survive this worst-case scenario, it is not strange that it can survive other defenses.
>
> Also from the empircal side, in Appendix B.8, we conducted a case study on Krum and looked into the reason why A3FL outperforms baselines. We recorded the ASRs corresponding to the training rounds, and highlight rounds that an attacker-compromised client is selected by the server. We have the following observation from Figure 14:
> * A3FL-compromised clients are not more frequently selected by the server.
> * Once selected, A3FL can achieve higher and more persistent attack ASR than any other baseline attacks, since A3FL can maintain higher ASR when transferred to the global model.
>
> Both the model design and empirical observation should be able to explain why A3FL can still achieve high attack performance on detection-based methods.
>
> > Q2: The impact of the number of available training data on the attack performance is not clear.
>
> We thank the reviewer for the insightful comment. To study the impact, we introduce **data resizing factor $\gamma$**, and assume that each client has only $1/\gamma$ private dataset compared to the default setting. We vary the value of $\gamma$ and record ASRs on FedAvg corresponding to the number of compromised clients (P). As observed in **Table 1 in the attached PDF**, the ASR is significantly impacted only when there is merely 1 compromised client and $\gamma$ is larger than 8. In other cases we do not observe obvious drop in ASR. Therefore we can conclude that attack performance is not sensitive w.r.t. the number of available training data.
>
>
> > Q3: Can you provide some directions regarding potential defense strategies?
>
> We thank the reviewer for the interesting question. A3FL is based on a typical FL setting where there is only one global model and currently we didn't find a perfect defense strategy yet. However, in some non-typical FL settings (e.g., if the server can maintain multiple global models trained by sampled clients and aggregate these models for evaluation), the chance for  attacker to successfully inject the trigger can be lower especially when the number of compromised clients is limited. However, this potential defense strategy could be much more computational expensive in comparison to typical FL algorithms.
>
> > Q4: The trigger pattern is limited to a rectangular shape. It would be interesting to explore the adoption of irregular shapes in the proposed attack.
>
> Thank you for your insightful suggestions. In this paper, we adopt the same trigger shape for each evaluated attack to ensure fair comparison, which is also well acknowledged in previous works[4][5]. The idea of A3FL does not rely on trigger shape, thus should be direclty applicable to other trigger designs. We will consider backdoor triggers encompassing the entire range of the image in our future work.
>
> > Q5: More datasets and architectures can be considered.
>
> We further evalute A3FL on FEMNIST dataset and VGG16 model. We record our results in Table 2 and 3 in the attached PDF, and observe that A3FL can still achieve high ASR.

---

> > ### Comment · Reviewer_i2mP · 2023-08-12
> >
> > Thank you very much for the detailed response.
> >
> > Overall I appreciate the comments. Most of my concerns are well addressed. After carefully reading other reviewers' comments and the authors' responses to them, I believe in the correctness of my evaluation despite the significant divergence with Reviewer k1k8. (Reviewer k1k8 put really valuable comments on related works, however, Reviewer k1k8's questions 5 and 6 which are more critical to the paper's contribution are well addressed by the author.)

---

> > > ### Author Response · Authors · 2023-08-13
> > >
> > > We sincerely thank the reviewer for the valuable feedback. Your constructive comments have been improving the quality of our work. We are also delighted to learn that most of your concerns have been addressed to your satisfaction.
> > >
> > > Once again, we appreciate your time and effort in reviewing our paper. Please let us know if you have any further suggestions or concerns.

---

### Official Review · Reviewer_iLct · 2023-07-25

**Soundness:** 3 good
**Presentation:** 3 good
**Contribution:** 3 good
**Rating:** 7
**Confidence:** 2

**Summary:**

This paper presents a new backdoor attack method termed A3FL to address some limitations of existing predetermined and fixed backdoor attack methods. The proposed method can adversarially adapt to the dynamic global model so that when transferring to the global model, the local-optimized trigger will be affected very little. The method is benchmarked against and performs at par with or better than a number of state-of-the-art methods.

**Strengths:**

- The paper presents an adversarially adaptive backdoor attacks to Federated Learning.
- A3FL can alleviate the problem of suboptimal attack performance caused by existing work ignoring the global training dynamics.
- The method is benchmarked against relevant recent work.

**Weaknesses:**

- In the abstract, it would be better to give the full name of A3FL.
- It is recommended to discuss the limitations of the proposed method in order to help other scholars improve it.

**Questions:**

Please refer to the "weakness" part.

**Limitations:**

No apparent limitations were found.

---

> ### Author Rebuttal · Authors · 2023-08-10
>
> ## reviewer iLct
> We thank the reviewer for the positive feedback and constructive comments on our work.
>
> >Q1: In the abstract, it would be better to give the full name of A3FL.
>
> Following your suggestion, to improve the clarity of our paper, we will include the full name of A3FL in the abstract,in the updated version.
>
> > Q2: It is recommended to discuss the limitations of the proposed method in order to help other scholars improve it.
>
> We thank the reviewer for insightful comment. Currently A3FL is conducted on the image tasks, and it would be interesting to the robustness of FL in other scenarios, such as large language models. What's more, it would be interesting to explore the application of A3FL in other FL scenarios, such as vertical FL. We will discuss the limitations in the future work section.

---

> > ### Comment · Reviewer_iLct · 2023-08-21
> >
> > Thanks for your rebuttal. Although I’m not an expert in this field, after reading the comments of other reviewers and the author’s replies, I think this paper remains a positive contribution to the community. Therefore, I tend to maintain my original score.

---

> > > ### Author Response · Authors · 2023-08-21
> > >
> > > We appreciate your feedback and comments in improving our paper. Welcome to share any additional suggestions you may have.

---

### Author Rebuttal · Authors · 2023-08-10

## General Response to All Reviewers

We thank all reviewers for your constructive suggestions and insightful questions! We have responded to them in our separate responses. We have provided supplementary experimental results about different available training data and the attack performance of A3FL on other mode architectures and more datasets. We also discuss the computational overhead of A3FL in comparison to other attacks. We evaluate A3FL against post-training as well. Welcome to raise following questions to further improve our work!

The citation list in our defense is as follows:

[1] Nguyen, A., & Tran, A. (2021). Wanet--imperceptible warping-based backdoor attack. arXiv preprint arXiv:2102.10369.

[2] Doan, K., Lao, Y., Zhao, W., & Li, P. (2021). Lira: Learnable, imperceptible and robust backdoor attacks. In Proceedings of the IEEE/CVF international conference on computer vision (pp. 11966-11976).

[3] Doan, K., Lao, Y., & Li, P. (2021). Backdoor attack with imperceptible input and latent modification. Advances in Neural Information Processing Systems, 34, 18944-18957.

[4] Fang, P., & Chen, J. (2023). On the Vulnerability of Backdoor Defenses for Federated Learning. arXiv preprint arXiv:2301.08170.

[5] Lyu, X., Han, Y., Wang, W., Liu, J., Wang, B., Liu, J., & Zhang, X. (2023, February). Poisoning with cerberus: stealthy and colluded backdoor attack against federated learning. In Thirty-Seventh AAAI Conference on Artificial Intelligence.

[6] Zhang, Z., Panda, A., Song, L., Yang, Y., Mahoney, M., Mittal, P., ... & Gonzalez, J. (2022, June). Neurotoxin: Durable backdoors in federated learning. In International Conference on Machine Learning (pp. 26429-26446). PMLR.

[7] Geiping, Jonas, et al. "Inverting gradients-how easy is it to break privacy in federated learning?." Advances in Neural Information Processing Systems 33 (2020): 16937-16947.

[8] Li, Henger, Xiaolin Sun, and Zizhan Zheng. "Learning to attack federated learning: A model-based reinforcement learning attack framework." Advances in Neural Information Processing Systems 35 (2022): 35007-35020.

[9] Salem, Ahmed, et al. "Dynamic backdoor attacks against machine learning models." 2022 IEEE 7th European Symposium on Security and Privacy (EuroS&P). IEEE, 2022. [10] Doan, Khoa D., Yingjie Lao, and Ping Li. "Marksman backdoor: Backdoor attacks with arbitrary target class." Advances in Neural Information Processing Systems 35 (2022): 38260-38273.

[11] Wang, Hang, et al. "Universal post-training backdoor detection." arXiv preprint arXiv:2205.06900 (2022).

[12] Wu, Chen, et al. "Mitigating backdoor attacks in federated learning." arXiv preprint arXiv:2011.01767 (2020).

[13]Nguyen, T. D., Rieger, P., De Viti, R., Chen, H., Brandenburg, B. B., Yalame, H., ... & Schneider, T. (2022). {FLAME}: Taming backdoors in federated learning. In 31st USENIX Security Symposium (USENIX Security 22) (pp. 1415-1432).

[14]Li, H., Ye, Q., Hu, H., Li, J., Wang, L., Fang, C., & Shi, J. (2023, May). 3DFed: Adaptive and Extensible Framework for Covert Backdoor Attack in Federated Learning. In 2023 IEEE Symposium on Security and Privacy (SP) (pp. 1893-1907). IEEE.

[15]Alam, M., Sarkar, E., & Maniatakos, M. (2022). PerDoor: Persistent Non-Uniform Backdoors in Federated Learning using Adversarial Perturbations. arXiv preprint arXiv:2205.13523.

---

> ### Comment · Area_Chair_WZnV · 2023-08-20
>
> Thanks to the authors for their rebuttals and engagement with the author discussion process. In some cases I realise the reviewers have not yet acknowledged these rebuttals - I'd like to assure the authors that the conference has reminded the reviewers of the need to fully engage with author rebuttals, and that there are still opportunities for this to happen during this (almost finished) author discussion period, and then into the reviewer discussion period that comes next. I'd also like to reassure the authors that I've read in full all of the reviews, the rebuttals and all comments to date. I'll focus mostly on those reviews yet to reply, and identified concerns/weaknesses (with strengths already noted elsewhere).
>
> **i2mP** has responded in the OpenReview thread to the rebuttal and follow-up. One initial concern of theirs - absence of defences in the attack's modelling - and the authors' framing of Eq 3 as worst case raises a follow-up. I understand that considering a (hypothetical) global model that unlearns the trigger is fruitful (shown empirical) is it really worst case in a theoretical sense? For example, an equilibrium strategy could be considered worst-case by being minimax in some way. I don't see that happening here. That's maybe fine, as it doesn't diminish the empirical results, but I'm not sure the language around worst-case is precise. Nor does it really address questions around modelling defenses in this case. (That is, for many questions the author response is "it's worst-case so we don't need to consider this". Similar commentary arises with **BFfe**, for example their W1 and the author response seems to acknowledge this concern.
>
> ### iLct
> **W1/W2)** thanks to the authors for responding and committing to address the reviewer's concerns. That said, the review is a little light on detail and I'd suggest (for example in addressing limitations) to also consider the feedback of the other reviewers who provide much more detail on limitations.
>
> ### k1k8
> _(adopting the authors' numbering system)_ **Q1 lines 36-37)** I appreciate the clarification of suboptimality of fixed-triggers (but is this paper considered optimal? Is this enough of a discussion if only focusing on suboptimality?). I appreciate that there are differences in how one uses "stealth". I appreciate that in this FL setting the server may not see images, however it is conceivable that an insider threat at a client party still wants to be stealthy in the conventional sense (imperceptible change). I think more nuance would be helpful, in both sides of the discussion. **Q2 L45-46)**  Thanks for this helpful discussion. However, as discussed, I'm not convinced about the claims of "worst case" analysis. **Q3/4)** Thankyou **Q5)** I think I understand that the client has the trigger through its "simulation", however a broader point is somewhat loose language around "unlearning". For example, see the literature on machine unlearning which specifically refers to a certifiable (provable) definition of unlearning. The method here is intuitive but I don't currently think there is true unlearning in the modelling, and like "worst-case" I worry language being imprecise might mislead the reader. **Q6/9)** thankyou. Regarding **Q7/8)** it is unreasonable for authors to consider a 3DFed paper published July 2023, unless there's an earlier arXiv. However the broader point of the reviewer is that backdooring is a huge active area with recent SOTA not examined. It isn't always possible to do such examination during the author discussion, and one can't usually compare against everything in existence, however it is reasonable for reviewers to ask for missing comparisons. I appreciate the updated results including on FLAME, thank you.

---

> > ### Comment · Reviewer_i2mP · 2023-08-21
> >
> > Thanks Area Chair WZnV for pointing out the "worst case" issue. I agree that the worst-case language is imprecise, since the "worst case" in the paper is an **approximate** of the actual worst case, as Review BFfe suggests, the actual worst case should include local training data of all benign clients.

---

> > ### Author Response · Authors · 2023-08-21
> >
> > We sincerely thank you for your insightful comments and suggestions on our paper. We found them particularly helpful in improving the clarity of our work and the preciseness of our language. We hope the following explanations would make it clear to you and other reviewers, and we will add them into our main paper in the revision.
> >
> > > Follow-up on **Reviewer i2mP Q1**, I understand that considering a (hypothetical) global model that unlearns the trigger is fruitful (shown empirical) is it really worst case in a theoretical sense? ... Nor does it really address questions around modelling defenses in this case. (That is, for many questions the author response is "it's worst-case so we don't need to consider this".
> >
> > Thank you for bringing this up and we think it is a good question. We acknowledge that Eq. (3) does not give the rigorous "worst-case" in a theoretical sense and as Reviewer i2mP commented, it is an approximation of the actual worst case, which is hard to achieve in real FL settings. We will revise the paper and change the statements about "worst-case" accordingly to avoid potential confusion. Going back to the question about defense modelling, intuitively, the adversarially crafted model $\theta_t'$ represents the ultimate goal of the defenders, i.e., mitigating the impact of the backdoor trigger. Essentially, that is the goal of all potential defenses (including unseen ones). Once the backdoor trigger can no longer impact the prediction results, the defender's goal is achieved. Therefore  Eq. (3) allows us to simulate this ultimate defender goal and be prepared to optimize a persistent backdoor trigger aiming to survive possible defenses.
> >
> >
> >
> >
> > >  Follow-up on **Reviewer k1k8 Q1**, I appreciate the clarification of suboptimality of fixed-triggers (but is this paper considered optimal? Is this enough of a discussion if only focusing on suboptimality?).
> >
> > We appreciate your comment and are sorry for the confusion. We agree that it is better to describe fixed-trigger attacks more comprehensively rather than just say 'sub-optimal'. We will modify the sentence to "While a fixed-trigger attack can be more efficient and straightforward, it usually suffers from limited effectiveness and more obvious utility drops."
> >
> >
> > > Follow-up on **Reviewer k1k8 Q1**, I appreciate that there are differences in how one uses "stealth". I appreciate that in this FL setting the server may not see images, however it is conceivable that an insider threat at a client party still wants to be stealthy in the conventional sense (imperceptible change). I think more nuance would be helpful, in both sides of the discussion.
> >
> > Thank you for your in-depth discussion on this and we agree that it is better to make this clear with more nuanced words. We would like to define the "stealthiness" in this paper as "**utility stealthiness**", i.e., the difference between the utility of unimpacted and backdoored models. We understand that data stealthiness, i.e., imperceptible change in image, also contributes to the general stealthiness of injected backdoor. In this paper, we adopt the same trigger constraint to ensure the same level of data stealthiness for all attack baselines for a fair comparison. We will discuss and clarify this in the revision.
> >
> > > Follow-up on **Reviewer k1k8 Q5**, I think I understand that the client has the trigger through its "simulation", however a broader point is somewhat loose language around "unlearning". For example, see the literature on machine unlearning which specifically refers to a certifiable (provable) definition of unlearning. The method here is intuitive but I don't currently think there is true unlearning in the modelling, and like "worst-case" I worry language being imprecise might mislead the reader.
> >
> > Thank you for pointing out and we are sorry for the confusion. In this paper, the "unlearning" refers to the "backdoor unlearning" which follows the backdoor unlearning literature [16] [17] in the context of backdoor attacks. In backdoor unlearning, the defender mitigates the impact of backdoor by minimizing the prediction loss produced by the backdoor trigger. We will clarify this in the revised paper as well as discuss the difference between backdoor unlearning and machine unlearning to alleviate possible misunderstandings.
> >
> >
> > Once again, we sincerely thank you for your great efforts in moderating the rebuttal process and highly appreciate your engagement. The citation list in our response is as follows. Please let us know if you have any further suggestions.
> >
> > [16] Zeng, Y., Chen, S., Park, W., Mao, Z. M., Jin, M., & Jia, R. (2021). Adversarial unlearning of backdoors via implicit hypergradient. arXiv preprint arXiv:2110.03735.
> >
> > [17] Wang, B., Yao, Y., Shan, S., Li, H., Viswanath, B., Zheng, H., & Zhao, B. Y. (2019, May). Neural cleanse: Identifying and mitigating backdoor attacks in neural networks. In 2019 IEEE Symposium on Security and Privacy (SP) (pp. 707-723). IEEE.

---

> > > ### Comment · Area_Chair_WZnV · 2023-08-21
> > >
> > > Thanks to the authors for these clarifications.

---

### Decision · Program_Chairs · 2023-09-21

**Decision:**

Accept (poster)

**Comment:**

Overall, I found the paper to have compelling empirical results with a sound methodology. While some of the "theoretical-style" arguments used to justify the core ideas need to be reworded (see below) I didn't read these as theory grounding for the work, and ultimately, the ideas are novel and there's sufficient evidence to demonstrate where they work and limitations. The idea of adapting the backdoor trigger to global training dynamics is well argued. Results demonstrate that the proposed adaptation A3FL does reduce the rate of backdoor removing during federated learning. The experiments are largely very comprehensive (a key strength of the work) with caveats below and in the discussion, demonstrating convincing results on ASR, utility, lifespan of triggers.

The paper's language around "worst-case" is quite problematic. These motivations have been shown during author discussion not to really be worst-case. While this doesn't invalidate actual results, the language is imprecise and potentially misleading - but as a presentation issue it can be easily addressed and I'm glad that the authors have committed to doing so.

Second, baseline attacks have been discussed during the author discussion phase. I still believe these would make the work more compelling and improve its impact. However I viewed these as "nice to haves" rather than "must haves" and didn't critically limit the significance of the work.

I'd strongly encourage the authors to take on board discussions, in improving their paper. For example, the discussion of stealth, being precise when saying "worst case" (or perhaps not saying it), comparison/discussion briefly on machine unlearning (I don't think you need to certify unlearning! But reference to that body of work/discussion of relevance might be helpful), and inclusion of the updated results during the discussion e.g., on FLAME.